# The Version of the Composition of the Mausoleum-Khanaka Khoja Ahmed Yassawi Main Facade in Turkestan

Konstantin Samoilov [1,*], Bolat Kuspangaliyev [1], Gaukhar Sadvokasova [2] and Aizhan Akhmedova [2]

1   T. Basenov Institute of Architecture and Civil Engineering, Satbayev University, 22 Satbayev Str., Almaty 050000, Kazakhstan
2   International Education Corporation, Kazakh Leading Academy of Architecture and Civil Engineering/KazGASA, 28 Ryskulbekov Str., Almaty 050043, Kazakhstan
*   Correspondence: samconiv@mail.ru; Tel.: +7-701-357-68-93

**Abstract:** The problem of preservation, and optimal demonstration of it to local residents and tourists of architectural monuments, is of constant scientific and public interest. Two concepts coexist in dialectical interaction: conservation to preserve the monument in the form in which it has come down to in our times, and restoration of the monument with the restoration of lost details. In each case, one or another decision is made, which finds both supporters and opponents. One of the aspects of this problem is the attitude to buildings that have long breaks in the history of conduct in their construction works. An interesting example of such a monument is the mausoleum-khanaka of Ahmed Yassawi in Turkestan, which remains unfinished. Given its importance for the self-determination of the culture of modern Kazakhstan, it seems appropriate to consider the planned design, which remains unrealized. For the first time, the article suggests, by way of discussion, several options proposed by the authors for solving the main facade of the mausoleum-khanaka and the shape of the central dome.

**Keywords:** a reconstruction; a restoration; the medieval architecture of Central Asia; mausoleum-khanaka; a graphic analysis of composition; proportions

## 1. Introduction

The Medieval architecture of Kazakhstan is represented by a variety of monuments. They originally reflected the specifics of each of the historical periods in which they appeared. The mausoleum-khanaka of Ahmed Yassawi in Turkestan is one of particular importance. This building, despite the fact that it remains in an unfinished state, has great historical significance. Additionally, now, it is also considered as part of the spectrum of elements of regional identity for the modern architecture of the country [1]. Special attention is being paid to it at the present time. This is due to the fact that the city of Turkestan has become the capital of the region with the same name. In addition, in ideological terms, this city with a complex of medieval monuments is perceived as the "Spiritual capital of Kazakhstan". The studies mentioned in the literature review reflect the main part of the spectrum of available works on the topic under consideration. The works of the largest experts on the topic, published in the first half of the twentieth century, and modern studies summarizing and clarifying the existing body of knowledge are mentioned here.

The mausoleum-khanaka in Turkestan has been described and studied in detail for more than one-hundred-and-fifty years. By the beginning of this century, the scientific basis for the study of the mausoleum-khanaka consisted of more than 50 names of research materials [2]. The study of the monument is actively continuing.

One of the first published descriptions of the building was made by M.S. Bekchurin on 19 February 1866 [3]. He pointed out the location of the building in the structure of the city fortress, and briefly described the spatial composition, planning solution, functional scheme, and constructive solution. A list of state and public figures buried in the complex is given. Much attention is paid to the description of the organizational and financial aspects

of the existence of the complex as an object of religious worship. The description contains data on the time of construction of the "Azret mosque"; the legend about the reasons that prompted Timur to order the construction of the complex is retold. Special attention is paid to the translation of inscriptions made in Arabic and Persian. Attention is also drawn to the emergency condition of some parts of the building. The destruction was the result of both military actions that took place at various periods and untimely routine repairs.

In 1929 and 1930, a detailed description was performed by M. E. Masson [4,5]. He spoke about Khoja Ahmed Yassavi himself; traced the history of the mausoleum and the main stages of its functioning; characterized the historical and architectural character of the mausoleum; gave a description of the architectural planning and spatial solutions of the mausoleum; and described in detail the modern appearance of the mausoleum. An assumption was made about the possible similarity in the composition of the unfinished portal of the Mausoleum in Turkestan with the portal of the Ak-Saray Palace complex in Shakhrisyabz. Special attention is paid to the sheikhs of the Mausoleum and the legends of Khoja Ahmed. These works by M. E. Masson are the first scientifically substantiated description of the monument, which became the basis for further research.

In 1950, the work of G. I. Patsevich was published [6]. It describes, in detail, the complexities of the work on the repair and restoration of the mausoleum-mosque of Khoja Ahmed Yasawi, which were carried out in 1939–1941. Thanks to this description, the specifics of the constructive solution of the building became clear, both in general and in individual nodes.

Based on previously personally conducted research and analysis of the available scientific base in 1960–1963, L. Yu. Man'kovskaya published several works on the architectural and archaeological aspects of the restoration of the mausoleum [7–9]. For the first time, the architectural features of this building became the subject of dissertation research. In the future, under the leadership of L. Yu. Man'kovskaya, work was carried out to prepare the restoration, and a project was carried out to preserve the mausoleum of Khoja Ahmed Yassawi (1968–1972). This work became the basis of a comprehensive study on the typology of architecture in Central Asia. The research carried out by L. Yu. Man'kovskaya contributed to the "clarification of the architectural phenomenon of the resolution of many functions in a single form", which revealed a kind of universal constructor, from which all types of monumental buildings were erected, while remaining specific and unique. L.Yu.Man'kovskaya was able to prove that there was a standard design. At the same time, variations were allowed on the basis of a typical solution. In addition, the architects possessed methods of flexible transformation of spatial structures. In a monograph published in 1980 [10], a typological classification was developed for the first time covering all types of human activity in medieval society and the types of buildings and structures corresponding to them: it is based on the leading feature of architectural form—the type of three-dimensional structure. The extensive factual material of many years of full-scale study of the possibility of studying the art of Central Asian architects was summarized.

At the end of the 1980s, B. T. Tuyakbayeva highlighted, at a dissertation level, the composition and content of the epigraphy of the mausoleum-khanaka [11]. This work gives the most complete description of the specifics of the architectural decoration of the monument to date.

The authors and work devoted to the direct analysis of various aspects of the form and decoration of this building are: N. B. Alliyar and K. I. Samoilov (the currently available material on the architectural and artistic solution of the mausoleum-khanaka is briefly summarized [12–14]), B. A. Baitanaev and Yu. A. Yelgin (clarified the issues of onomastics and epigraphy of the mausoleum-khanaka [15]), T. K. Basenov (basic information on the architecture of the mausoleum-khanaka is given [16]), D. Mustapayeva (the supposed features of the original mausoleum of the twelfth century on the site of the mausoleum-khanaka of the fourteenth century are described [17]—this work is important for understanding

the history of the complex, which is still unfinished), N.-B. Nurmukhammedov (a somewhat romanticized description of the complex and the history of its development [18]), A. Ordabaev (the general issues of the study and restoration of the mausoleum-khanaka are described [19]—the interview describes the personal experience of many years of work on the study and restoration of the monument), Yu. A. Yelgin (the materials on the study and restoration of the mausoleum-khanaka from the mid-nineteenth century to the mid-twentieth century are summarized [20,21]—these studies form the basis for further restoration and reconstruction work), among others. These studies, conducted at different times, emphasize many aspects of the protection, study, repair, and restoration of the mausoleum-khanaka, which are important for the study of the authors of this article.

Considerations of this building in the context of the contemporary architecture of the Central Asian region is reflected in these studies: N. B. Alliyar and K. I. Samoilov (the general features of the historical and architectural context of the construction of the mausoleum-khanaka in Turkestan are considered [22]), E. Baitenov (the specifics of shaping in the memorial architecture of Kazakhstan as a whole are emphasized [23]), E. Baitenov, A. Tuyakayeva, and G. Abdrassilova (the processes of origin and development of various forms of Kazakh mausoleums are detailed [24]), E. G. Barsukova (the specifics of Sufism from the point of view of architectural and artistic solutions of religious buildings in medieval Central Asia are analyzed [25,26]), S. Chmelnizkij (the general characteristic of Timurid architecture as a historical and cultural phenomenon is given [27]), B. Glaudinov (the processes in the architecture of Kazakhstan at all stages of its ancient and medieval development are analyzed [28]), M. B. Kasymbekova, B. A. Glaudinov, and K. I. Samoilov (the specifics of Islamic memorials and religious buildings in medieval Kazakhstan are considered [29]), M. B. Kozha (a number of positions on the shaping of medieval monuments of Turkestan, obtained on the basis of new research, have been clarified [30]), A. Kh. Margulan, T. K. Basenov, and M. M. Mendikulov (the architecture of Kazakhstan before the beginning of the twentieth century is comprehensively considered [31]), G. A. Pugachenkova (the specifics of the architecture of the region in the fifteenth century are specified [32]), G. A. Pugachenkova and L. I. Rempel' (the specifics of the architecture in the context of the art of the region are shown [33]), D. J. Roxburgh (an overview of materials on architecture and art of the epoch is given [34]), K. I. Samoilov (the monuments of the memorial architecture of the Middle Ages are considered in the context of the centuries-old architecture of Kazakhstan [35]), A. Tondi (the architectural decor in the monuments of the epoch is investigated [36]), I. I. Umnyakov (the specifics of the research and restorative works of architectural monuments carried out in the 1920s in the region are shown [37]), V. L. Vyatkin (the ancient Samarkand monuments are described in detail [38]), B. V. Weimarn (a general overview of architecture in the context of the art of the region is given [39]), A. Ya. Yakubovsky (the construction activity of Emir Timur in Samarkand is described [40]), P. Zahidov (based on studies of Timurid architecture, conclusions are drawn about the presence of a peculiar system of proportionation [41–43]), B. N. Zasypkin (a general description of the architecture of the region is given, and the features of the most significant structures are specified [44]), among others (general information on the architecture of most of the medieval monuments preserved in the region is given [45,46]). In these works, an understanding of the processes of development of the architecture of the region is given from the perspective of the knowledge available during the period of the study. The body of knowledge is constantly expanding, and some of the provisions presented in the studies of the middle of the last century have now been supplemented and partially corrected. All these studies are important for understanding the process of formation of the modern appearance of the mausoleum-khanaka.

From the point of view of the assumption of the original design, which was not fully realized, it is important to study the identical buildings of that era: N. V. Gilmanova (features of the Ak-Sarai Palace [47]), S. N. Kabanov (description of the preserved ruins

of the Ak-Sarai Palace [48]), Z. Khakimov (description of a number of buildings of the Timur era [49]), T. Kuziev (the specifics of the architectural composition of minarets in the region [50]), M. E. Masson (features of minarets in the context of the architecture of the Bibi-Khanim Mosque [51,52]), M. E. Masson, G. A. Pugachenkova and H. T. Sultanov (the development of the city of Shakhrisabz in its heyday [53,54]), L. Yu. Man'kovskaya and E. Paskaleva (some features of the Bibi-Khanim mosque [55–57]), among others (for example, the hidden meanings of shaping the Bibi-Khanim Mosque [58]). This is especially true of compositional analyses of the formation of various monuments of the region in general (M. S. Bulatov—the specifics of harmonization techniques in the architecture of Central Asia from the tenth–fifteenth centuries [59]), and individual examples in particular (Sh. E. Ratiya—features of the multidimensional study and restoration of the Bibi-Khanim mosque [60]).

Almost from the very beginning of the active study of the monument, there was an interest in the reconstruction of the original design of the structure, which is still in an unfinished state. However, apart from general indications of possible similarities to some modern buildings (Ak-Saray Palace in Shakhrisyabz, the main mosque of Bibi-Khanim in Samarkand), there were practically no attempts to, at least, graphically reconstruct the unfinished facade. The restoration works carried out since the middle of the twentieth century were focused on the restoration of partially destroyed cladding and the preservation of the monument. In this regard, it seems appropriate to conduct a kind of restoration experiment to recreate (at least graphically) the unrealized architectural and artistic design of the general appearance of the mausoleum-khanaka of Ahmed Yassawi in Turkestan. Additionally, there are not many questions about the decoration of the portal part of the mausoleum, since there are preserved fragments. Their interpretation, which is quite adequate from a historical and cultural point of view, is not particularly difficult. Determining the designed height of the portal and its flanking towers is very controversial. There are questions about the main dome. The definition of the form of the wedding of corner minarets is equally problematic. The geometric construction based on the compositional solution of similar structures in the region is methodically justified.

## 2. Materials and Methods

A common method of sequential data collection, analysis, and generalization was used to conduct this study. Initially, the differential method of studying the problem distinguishes examples with similar architectural forms from the aggregate. Next, the integral method groups selected examples according to the specifics of shaping. Continuing the study, the formal method examines the evolution of various architectural forms. Then, the iconographic method emphasizes the presence of certain features peculiar to the prototypes of the studied forms. The final analysis of the structural-semiotic method models the prospects for the evolution of architectural forms. The analysis methodology used is based on the principle of analogy. The principle of analogy is justified in this case since there is a triad—one historically short period, one region, and one customer. It is difficult to assume that in this combination of factors a cardinal difference in the principles of shaping is possible. The historical restoration of the beginning of the last century was based on the fantasy of the "ideal forms" of the epoch. In the situation under consideration, it is proposed to use standard shaping techniques that have been repeatedly implemented in buildings of similar size and function. At the beginning of the study, examples of construction completion in forms different from the original ones were mentioned. This "modernization" is seen by the authors as inadequate for the object under consideration. The proposed option suggests the possibility of reproducing the original design of the mausoleum-khanaka of Ahmed Yassawi. This approach is seen by the authors as the most attractive from a historical and educational point of view. It shows the greatness of the ancestors' plan. This

is very important for the modern outlook of the country's citizens. This demonstrates the deep historical roots of the country's modern culture. The use of modern architectonic language in this case seems unacceptable to the authors. The goal is not just to complete the construction, the goal is to reproduce the original design.

A fairly common feature of a certain number of buildings is that their construction stretches over a considerable period of time. Most of them are still being completed preserving the original idea and function. For example, the Cologne Cathedral (50°56′28″ N, 6°57′24″ E) was built, almost completely preserving the original design, from 1248 to 1880. The Redemptive Church of the Holy Family (Temple Expiatori de la Sagrada Família) in Barcelona (41°24′13″ N, 2°10′28″ E) continues to be built according to the project of Antoni Plàcid Guillem Gaudí i Cornet since 1883. The main stages of this building construction are shown in Figure 1.

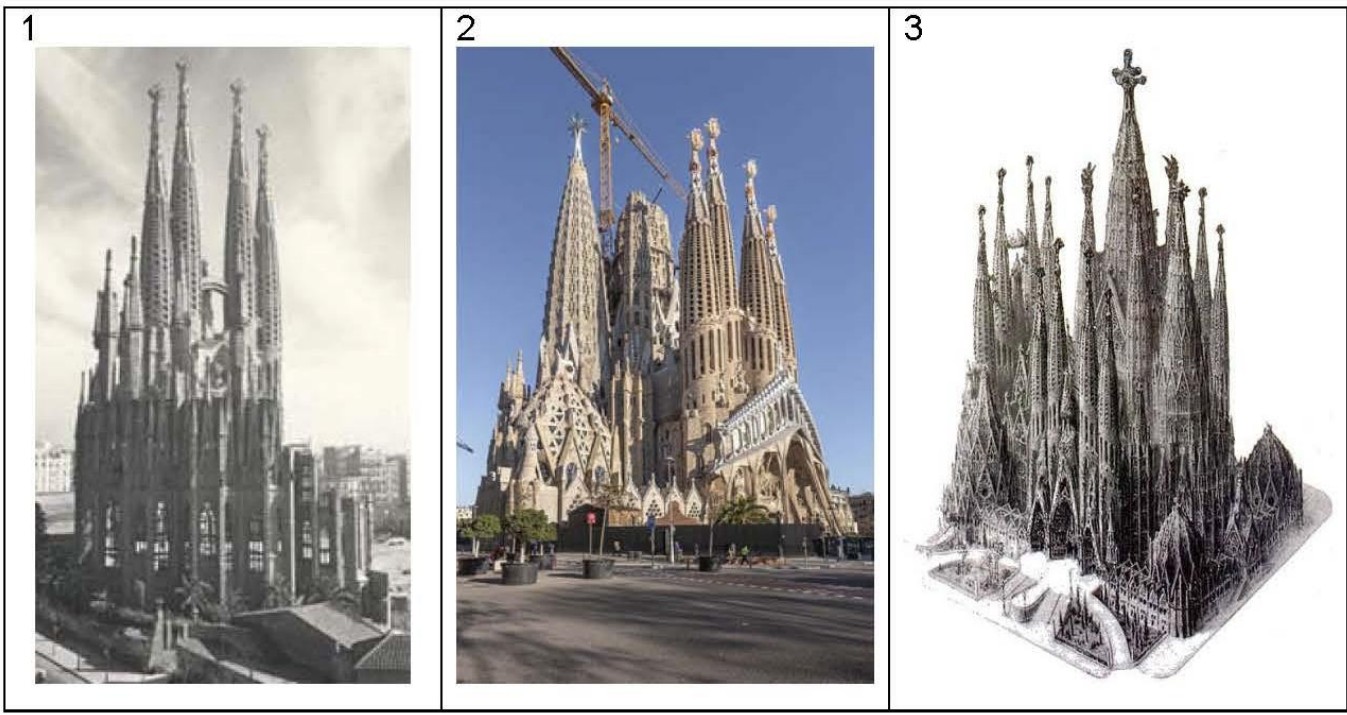

**Figure 1.** The Basílica i Temple Expiatori de la Sagrada Família, Barcelona (Francisco de Paula del Villar y Lozano, from 1877; Antoni Gaudí i Cornet, from 1883): 1—1953 [61]; 2—2021 [62]; 3—The model (for the ending in 2026 or late) [63].

Some buildings are being completed with changes that meet modern needs both in terms of function and imaginative solutions. St. Stephen's Cathedral in Vienna (48°12′30″ N, 16°22′23″ E) was started in 1137, went through several stages of rebuilding and expansion, and acquired a modern look by 1511. The building of the Cathedral of Santa Maria del Fiore in Florence (43°46′23″ N, 11°15′23″ E) began in 1296 according to the project of Arnolfo di Cambio. For several decades, the works were conducted under the direction of Giotto di Bondone, Francesco Talenti, Giovanni di Lapo Ghini, Andrea Pisano, Alberto Arnoldi, Giovanni d'Ambrogio, Neri di Fioravante, andAndrea Orcagna. These architects made some additions and changes to the original design. The construction was completed in 1436, when an innovative dome designed by Filippo Brunelleschi was erected for that era. The architectural and artistic solution of this dome, according to contemporaries, did not harmonize with the appearance of the main part of the cathedral. The later additions to the Cathedral of Our Lady of Chartres (48°26′50″ N, 1°29′16″ E), with towers of different shapes on the western facade (south–1140/1155, north 1513), look very contrasting. A similar

contrast is seen in the Cathedral of Saint Etienne in Bourges (47°04′55″ N, 2°23′58″ E). Examples of construction completion in a modified form are shown in Figure 2.

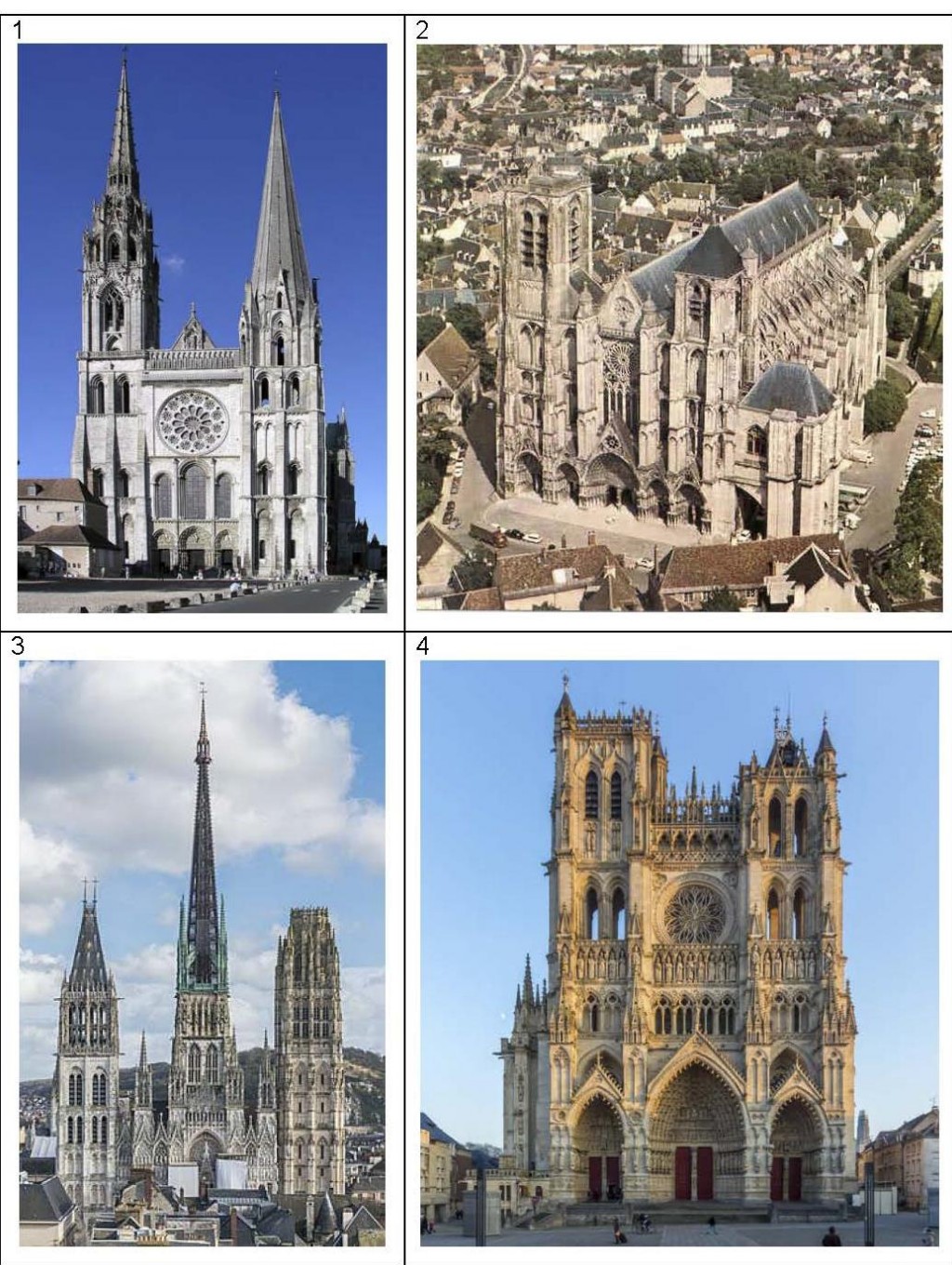

**Figure 2.** Examples of construction completion in a modified form: 1—The Cathedral of Our Lady of Chartres, 1260 (Towers–1140/1155)—The North Tower, 1513 (Build. Jehan de Beauce), Chartres, France [64]; 2—The Cathédrale Saint-Étienne de Bourges, 1324—The North tower, 1542, Bourdes, France [65]; 3—The Cathédrale primatiale Notre-Dame de l'Assomption de Rouen, 1506—The Northern tower (the Saint-Romain tower), 1145—The Southern tower (the Butter tower), 1485–1769–1884–1956–1999, Rouen, France [66]; 4—The Cathédrale Notre-Dame d'Amiens, 1269 (Arch. Robert of Luzarches, Thomas de Cormont, Renaud de Cormont),—the South tower, 1372; The North tower, 1420, Amiens, France [67].

The mausoleum-mosque of Arystan-Baba in the Kogam village in the Turkestan region (42°51′05″ N, 68°16′05″ E) has repeatedly changed its appearance. The original construction dates back to the XII century; the first reconstruction was made in the XIV–XV centuries; the second in the XVIII century; the third in 1909; and, due to the threat of destruction, the building was demolished and rebuilt in 1971 [46]. Large towers with tall lanterns are located at the corners of the main facade. The wide frieze, highlighted by cornices with belts of figured masonry, is dissected by blades with panels between them. The main entrance is accentuated by a pointed arch with large detailed clinched archivolt masonry. The wedding is made in the form of a developed pediment of a keel shape and framed by turrets. The arch is flanked by buttresses topped with wall-mounted pinnacles with pointed domes. On the front face of the pylons there is a panel with a medallion in the middle. The large domes of the building have a pointed shape. One of them is crowned with a figured spire. The arched windows and door openings are framed by platbands with broken locks. The authors of this article suggest that the keeled shape of the pediment appeared only in the XVIII century, as a metaphor for the unfinished portal of the grand mausoleum-khanaka of Ahmed Yassawi in the XV century. The reason for it could be that the preacher Arystan Bab was the spiritual mentor of Khoja Ahmed Yassawi.

One of the earliest structures in Kazakhstan, which used curly brickwork in the form of fir trees, triangles, and rhombuses, is the mausoleum of Karakhan (42°54′02″ N, 71°23′14′E) in Taraz. The building dates back to the XI century. The mausoleum has a developed three-part portal with two turrets, not centered in the area of the walls of the portal proper. The portal is solved with figured brickwork. The masonry has several varieties with the use of both terracotta and glazed tiles. By the beginning of the twentieth century, the building began to need major repairs. After reconstruction in 1905—1906, the mausoleum of Karakhan has completely lost its original appearance. As A. S. Schenkov points out, restructuring in the modern forms at that time is explained by the peculiarity of the worldview, which put the sacred function of the structure first, which should continue to function while maintaining syncretic integrity. This determined the primacy of integrity during restoration "to the detriment of the authenticity of the monument" [68] (p. 178). In fact, the completely new building has a developed complexly dissected portal with blind corner towers topped with spheroconic domes on three-part cornices. A large lancet niche with a cantilever impost is placed in the shallow central two-part rectangular niche of the portal. Symmetrically from this niche, shallow two-part pointed and rectangular niches are arranged in three tiers. The side and rear facades have deep lancet niches along the axis, inscribed in rectangular shallow niches, which, in turn, are flanked by high lancet niches. A large spheroconic dome is placed on a low quad. From the construction that reached the beginning of the twentieth century, after the reconstruction, only the theme of the three-part division of the facades remained in appearance [35,69].

The mausoleum of Aisha-Bibi (42°50′01″ N, 71°12′37″ E) demonstrates an interesting interpretation of the architectural and artistic ideas of that period. The mausoleum is located in the village, which now has the name Aisha-Bibi and administratively belongs to the Dzhambul region. The building dates back approximately to the XI century. The corner columns of the original shape (high truncated cones articulated through the torus), are almost all continuous (with elements of figured masonry) facing with terracotta ornamented tiles (20 types, 4 types of themes), with a wide variety of compositional connections, figured capitals of columns under arches, ornamented tympanums, and rounded shaped tiles when moving from edges to smooth walls to form a unique a solution that later became widespread in the region [29]. Considering the Central Asian monuments of the XI–XII centuries, B. N. Zasypkin points out: "Facing with terracotta tiles attached to brickwork has opened a new page in architecture. (...) A sharp line is established between the structural brick body and decorative and ornamental cladding, which was especially pronounced in the architecture of the XIV century" [44] (p. 50). By the beginning of the twentieth century,

a significant part of the wall of the western facade and fragments of other parts remained in place [68,70]. In 2002, under the leadership of N. Rameto, the building was restored. There are no questions about the shape and lining of the walls, since they are recreated on the basis of preserved fragments. The question of the smooth cone-shaped shape of the wedding and the polygonal drum under it remains debatable.

There are suggestions that this element could have large folds by analogy with the Babaji-Khatun mausoleum located next door (42°50′02″ N, 71°12′38″ E). This mausoleum, built a little later, is topped with a sixteen-rib cone. According to the well-preserved fragments, this element was restored in 1981. The cube-shaped building with a poorly developed portal and a ribbed conical dome on a sixteen-sided prismatic drum was plastically solved on a combination of differently recessed arched niches, simple round rosettes, frames, and toothed belts. The recessed planes of niches and frames are filled with ornamented tiles. A flat relief inscription on terracotta tiles is included in the low parapet. All facades, except for the smooth rear, have the same plastic solution. The building demonstrates the early stage of development of portal composition in the Region.

As an option for the reconstruction of the Aisha-Bibi mausoleum, a twelve–sided tent on a twelve-sided prismatic drum is possible—this is how the Mausoleum of Il-Arslan (Fahreddin Razi), built in the second half of the XII century in Kuneurgench (42°18′16″ N, 59°08′44″ E), was solved. In the analysis of the composition of the mausoleum, which was performed by M. S. Bulatov [59] (p. 111), the dome is depicted as spheroconic. The changes in the appearance of the Karakhan mausoleum and the Aisha-Bibi mausoleum are shown in Figure 3.

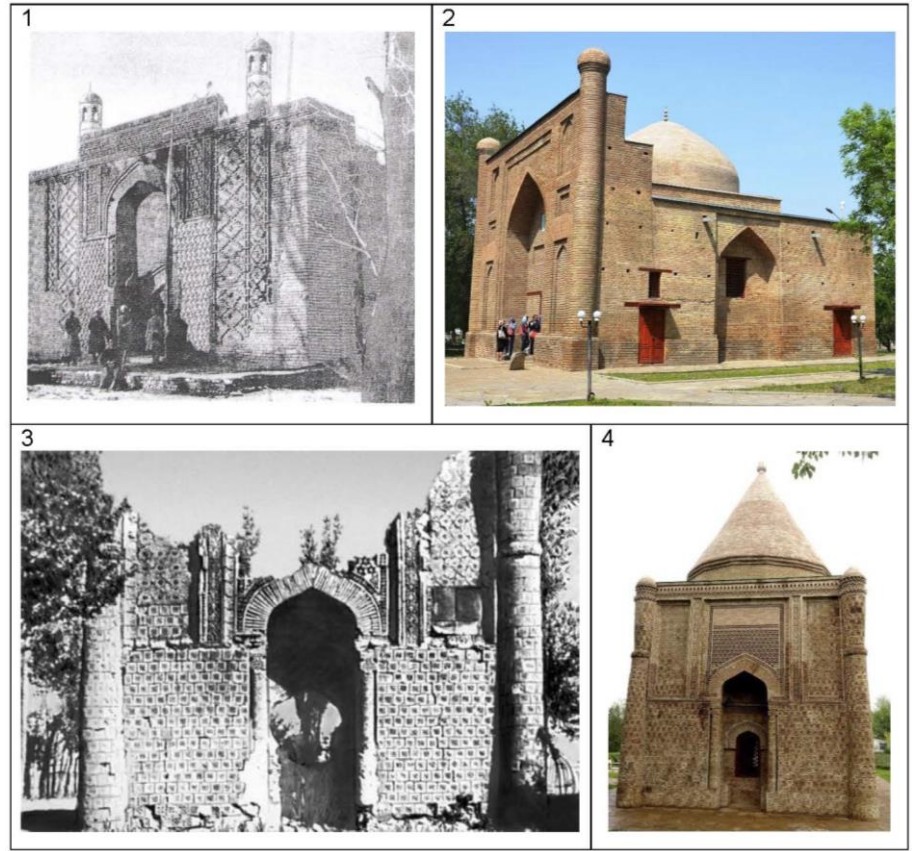

**Figure 3.** Examples of the reconstruction of medieval mausoleums in Kazakhstan: 1, 2—The Karakhan's Mausoleum, Taraz: the view in 1902 [35] (p. 25), the view after reconstruction in 1906 [71]; 3, 4—The Aisha-Bibi's Mausoleum, Aisha-Bibi village, Dzhambul region: the view in the middle of the twentieth century [35] (p. 27), the view after reconstruction at the beginning of the XXI century [72].

These examples show that the expediency of further use of a particular historical monument not only as a museum exhibit, but also as a part of modern life actively exploited for direct or indirect purposes, has deep historical roots. Completion, rebuilding, and adaptation are objective needs of vital activity. The degree of transformation of a historical monument depends on specific needs and ideological attitudes [35] (p. 849), [73]. These processes are present in all periods of the development of civilization. The current period of the evolution of the architectural and spatial environment is no exception.

Some buildings, now regarded as valuable monuments of the Central Asian Middle Ages, remained unfinished or were badly destroyed. This is especially true of the architecture of the Timur era. One of such grandiose structures is the mausoleum-khanaka of Ahmed Yassawi (43°17′52″ N, 68°16′15″ E). The mausoleum-khanaka of Khoja Ahmed Yasawi is the only construction undertaking of the emir, undertaken by him outside his hometown of Shakhrisyabz and the capital of his empire the city of Samarkand. The building is located in the fortress city of Yassy, which, from the middle of the XVI century, became known as Turkestan. The decree from the founder of the Empire defined the main parameters of the building and some details of its decoration. The construction of the building was carried out by Mawlyan Ubaidullah Said, Khoja Hussein al Shirazi, and Shelms Abd-al-Wahab al Shirazi. The construction took place in two stages in 1395—1399, and in 1591 partial completion and repair were carried out. In particular, the arch of the main facade was closed. However, the top of the arch was partially displaced relative to the central axis. In addition, the right part of the arch was partially deformed (arrow in Figures 4 and 6). The width of the portal (along the axes of the corner minarets) is 45.96 m. The pylons of the northern portal include the remains of a structure of the XII century, which had a glazed finish of predominantly green in combination with unglazed tiles [45]. The facades, almost devoid of fine plastic elements (except for the remaining unfinished portal part), have polychrome cladding made of glazed bricks. The remaining open square and rectangular niches in the lower part of the main portal and octagonal minarets were intended for the installation of mosaic panels. As pointed out by Sh. E. Ratiya [58] (p. 95), on the front side of this portal, the principle of dividing it into seven separate parts is clearly revealed; the equality between which, however, is violated here. The researcher admitted that with the timely completion of the construction and cladding of facades according to the plans of the architects who started the construction, this violation would not have been observed. It is possible that the inequality between the individual main divisions of the plan of the facade plane is caused only by the absence of cladding on its walls.

The question arises about the reasons for the cessation of work by the masters in 1399. The only work that remained was the completion of only the upper part of the portal and the cladding. Was it due to the need to start the construction of the Samarkand Mosque complex? Conversely, Emir Timur, according to the descriptions of his contemporaries, had great respect for the memory of Ahmed Yassawi. Accordingly, it is impossible to imagine that the building of the mausoleum-khanaka was halted only due to the transfer of all of the masters to another project. Moreover, the portal arch, the most important semantic element of the main facade, turned out to be incomplete. Additionally, judging by the composition and shape, the main dome is not completed. The termination of funding after the end of a successful military campaign in India also cannot be considered as an argument. The bulk of both the construction and finishing work was already completed. Given the large construction works that were carried out in Samarkand at the time, it is impossible to assume the complexities of financing an almost completely finished object in Turkestan. Maybe the Emir planned to finish the construction of the mausoleum-khanaka after the completion of the main works in Samarkand. This theory arises if you look at the intended appearance of the facades (Figures 15, 16, 17 and 18). The diagrams show the insufficient (from the point of view of composition) height of the main dome. This issue will be considered later. In general, the reason for the termination of the main construction work on the mausoleum-khanaka during the life of Emir Timur has not yet been clarified.

The portal, when completed in the XVI century, was crowned with teeth. These battlements served as a shelter for shooters in case of building defense or for sentries who notified residents of the enemy's approach (a large flag was hung out, and residents working in the fields hid inside the city walls). The arch was connected so as to form an unbroken protective line along the wall. The height of the portal with teeth has become about 30 meters. The basis of the decoration of the surfaces of the walls, drums, and domes is epigraphic [17,25]. The volumetric-plastic solution emphasizes the polyfunctionality of the structure. For example, the semantic importance of the mausoleum is emphasized by the allocation of its dome. This dome, unlike the main one, has a corrugated outer surface and a relatively high cylindrical drum. The main dome, being the largest in Central Asia (diameter 18.2 meters inside and 22.0 meters outside), is built on an octagon resting on a quad with beveled corners. The top of the dome is located at a height of 44 meters (inside the hall the height is 38.0 meters). According to the authors, the preserved shape of the main dome corresponds to the inner dome characteristics of the epoch. Accordingly, the authors assume that the construction of an external dome was originally envisaged (this assumption will be clarified further). Each individual volume of the structure has a clearly identified mosaic decoration with a three-part division. There is a functional differentiation of epigraphic decoration. In the appearance of the building, the sites erected in different periods are quite clearly traced.

Paradoxically, in the presence of an optimal design of walls and ceilings, the foundation of the building clearly does not correspond to the logic of construction; the depth of laying is only 0.3 m. The foundation itself is made of several rows of bricks. Clay layers were also made to a depth of 1.5 m. This became critical when, in 1846, the building was partially flooded with water for several months. As a result, the rate of destruction of the individual parts of the building has increased. In the early 1870s, the building was threatened with targeted destruction due to the fear that its cracked walls might collapse. The awareness of the cultural and historical value of the monument precluded its demolition and stimulated the activation of repair work.

In 1872, repairs were carried out, the main result of which was the cleaning of accumulated debris from the building. In 1884, the roof was repaired, spillways were arranged, collapsed sections of brickwork were laid in the lower part of the walls, and the lining of the outer surface of the main dome was started. As a result of the measurement work, the plan of the building was drawn. The ground movements that arose for various reasons led to the need to strengthen some sections of the southwestern and northwestern walls with buttresses (1886—1887). A detailed survey of the building began in 1905. Plans, sections, and facades of the building were drawn. In 1910, a partial re-laying of part of the walls and ceilings in the kitchen was carried out. Since 1922, regular inspections of the monument began to be carried out with a photo recording of the condition of the building as a whole and its individual parts. The inspection revealed multiple cracks, damage, destruction of arches in many parts of the building, and subsidence of the northwest corner. In 1925, major repairs of the building were also carried out: the lower parts of some rooms were shifted, the kudukhana dome was blocked, and the foundations were opened. The epigraphy of the mausoleum continued to be read.

In 1928, the lower parts of the walls inside the building and the main dome were repaired. In 1952, work began on the construction of concrete foundations under separate sections of the walls. The work was completed in the early 2000s. By the middle of the twentieth century, the building had lost some of the cladding on the walls and domes. This cladding was almost completely restored by the end of the century. To strengthen the lower part of the portal arch, steel cables were placed. In 1997, a well filled in, in the Middle Ages (diameter 0.9–1.0 m), was discovered in the geometric center of the square of the central room. Accordingly, a confirmed assumption was made that a twelfth-century complex existed at this place until the middle of the fourteenth century. The semantic center of this complex was likely a small mausoleum of Khoja Ahmed. In front of him was a vast courtyard with a well.

This expanded the understanding that appeared in the middle of the twentieth century that: "(1) a new building was not built entirely, before that there was another building with an inner courtyard (or a group of buildings forming a courtyard), later blocked by the dome of kazandyk; moreover, the mosque has older walls (the North-Western corner of the complex); (2) the main southern portal with minarets is attached to a previously existing building; for the installation of the dome the walls of the former courtyard were reinforced with linings, and the corridors-passages in these walls were narrowed (this explained the lack of foundations under the southern portal)" [74] (p. 146). That is, Emir Timur gave an order to build a portal and erect a dome over the courtyard of the existing complex.

The state of the monument at the beginning of the twentieth century and its modern appearance are shown in Figures 4–6.

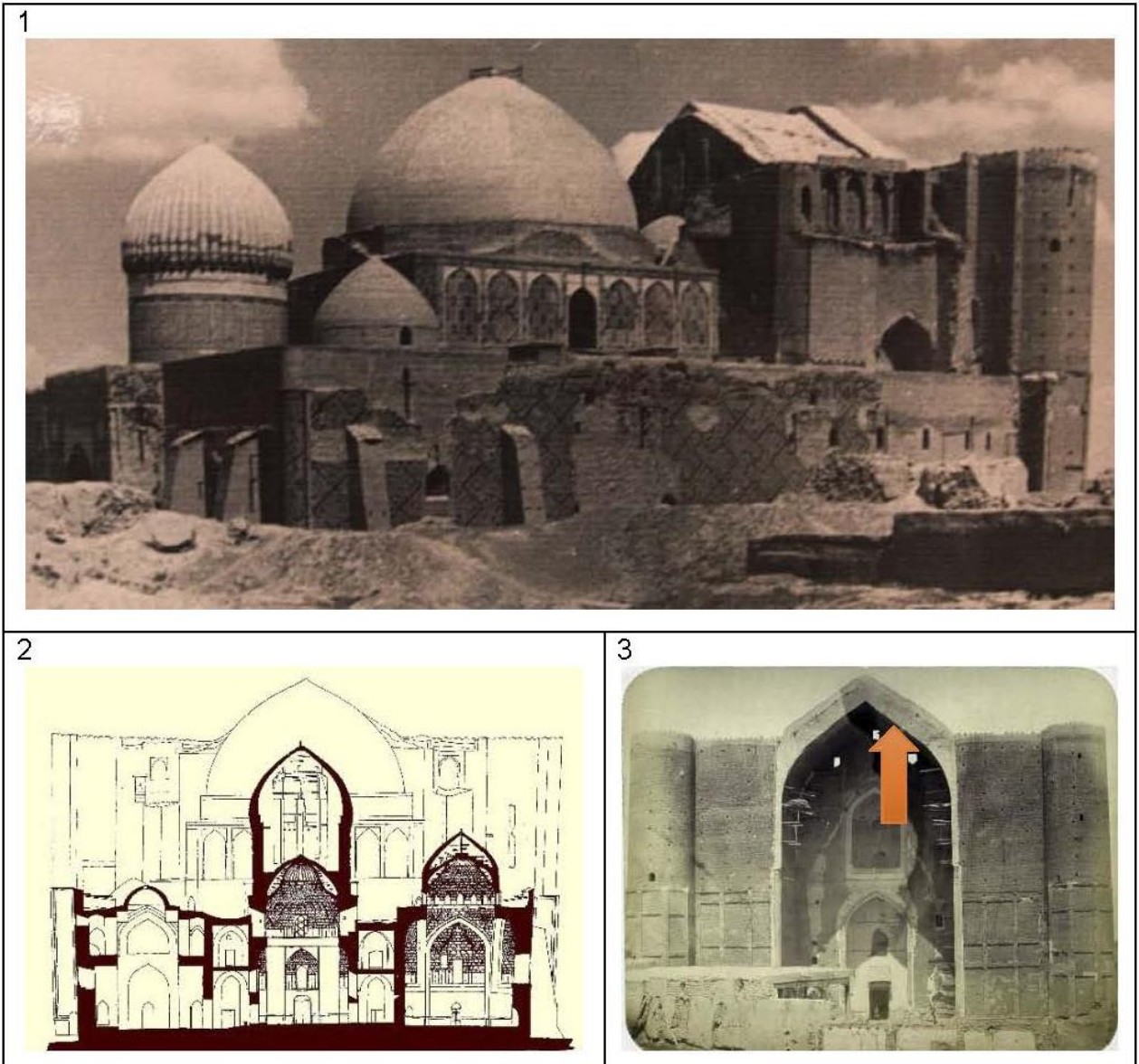

**Figure 4.** The mausoleum-khanaka Ahmed Yassawi, Turkestan: 1—The view in the first half of the twentieth century [75]; 2—The cross section [76]; 3—The southeastern facade at the beginning of the twentieth century [77] (the arrow shows the archivolt deflection).

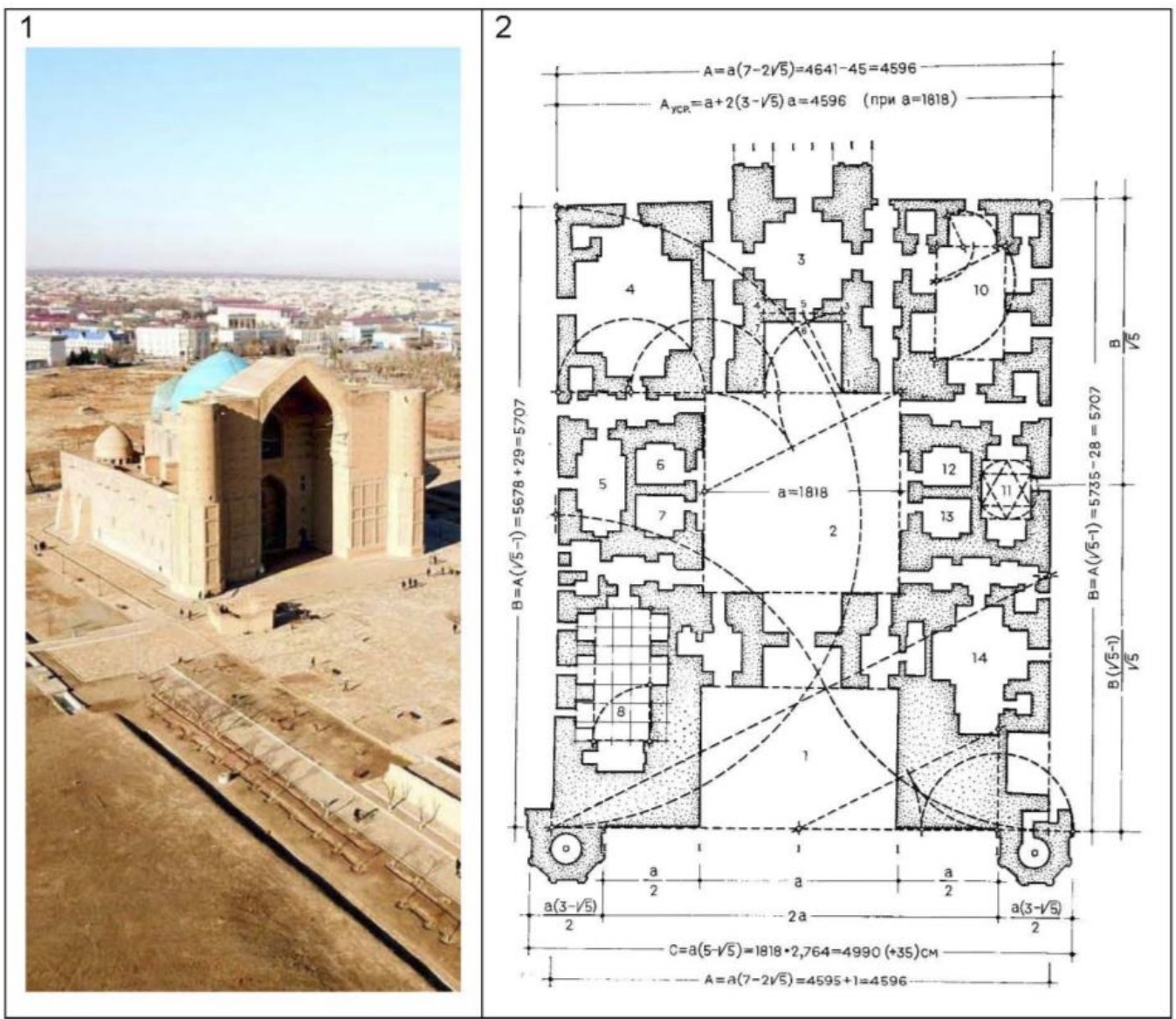

**Figure 5.** The mausoleum-khanaka of Ahmed Yassawi, Turkestan: 1—The general view in the middle of the twentieth century [78]; 2—The first level plan compositional analysis (M. S. Bulatov) [59] (p. 154).

For further research, it is advisable to consider the other buildings of the era. Their general composition and individual elements are important for determining the originally intended form of mausoleum-khanaka in Turkestan.

The Guri-Emir complex in Samarkand (39°38′54″ N, 66°58′08″ E) remained in an unfinished state. The main building was built in 1403—1404. It was included in the existing complex with a mosque and khanaka of Muhammad Sultan. Originally built octagonal in plan, the building is topped with a ribbed dome on a cylindrical drum (architect Muhammad ibn Mahmud Isfagani). The diameter of the dome is 15 m. In the future, various extensions will be made. The complex has a fenced yard from the north. The corners of the courtyard were fixed by four minarets. By the middle of the XIX century, only two minarets remained, adjacent directly to the main building. The northeast minaret collapsed in 1868, and the northwest minaret collapsed in 1903—they were restored in 1996. In the context of the problem considered in this article, the portal from the western side is of interest. This element remained unfinished; the pointed arch, brought to the height of

the middle of the dome drum, rises above the unfinished walls on the sides. Part of the wall collapsed. The builders' plans clearly included the creation of a much larger portal than the one located on the north side. The northern portal forms the main entrance from the front yard. In the future, the restoration work that has been going on since the XVII century may include the completion of the western portal as well as the restoration of the madrasah and khanaka. Accordingly, the cladding with ceramic panels can be made using historical analogue techniques. However, the demonstrated options for the reconstruction of the complex, for some reason, do not imply the completion of the construction of the great western portal. However, without this, the complex cannot appear in all its glory, demonstrating the greatness of the plans of the architects of the Timur era. Figure 7 shows the plan of the Guri-Emir complex, an unfinished portal on the west side, as well as a reconstruction option in the form of a model.

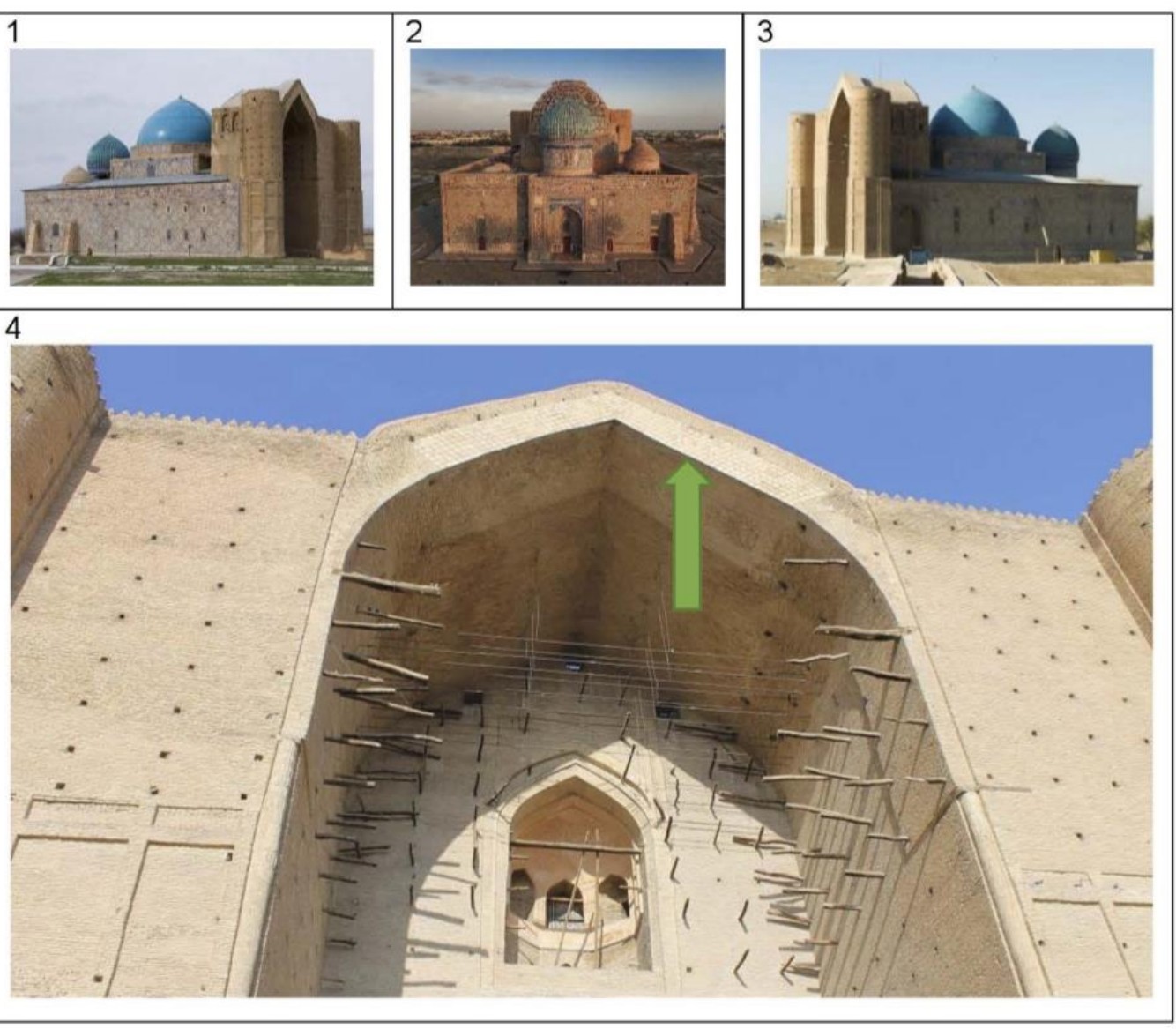

**Figure 6.** The modern view of the mausoleum-khanaka Ahmed Yassawi, Turkestan: 1—The view from the south [79]; 2—The view from the northwest [80]; 3—The view from the north [81]; 4—The fragment of the main facade [82] (the arrow shows the archivolt deflection).

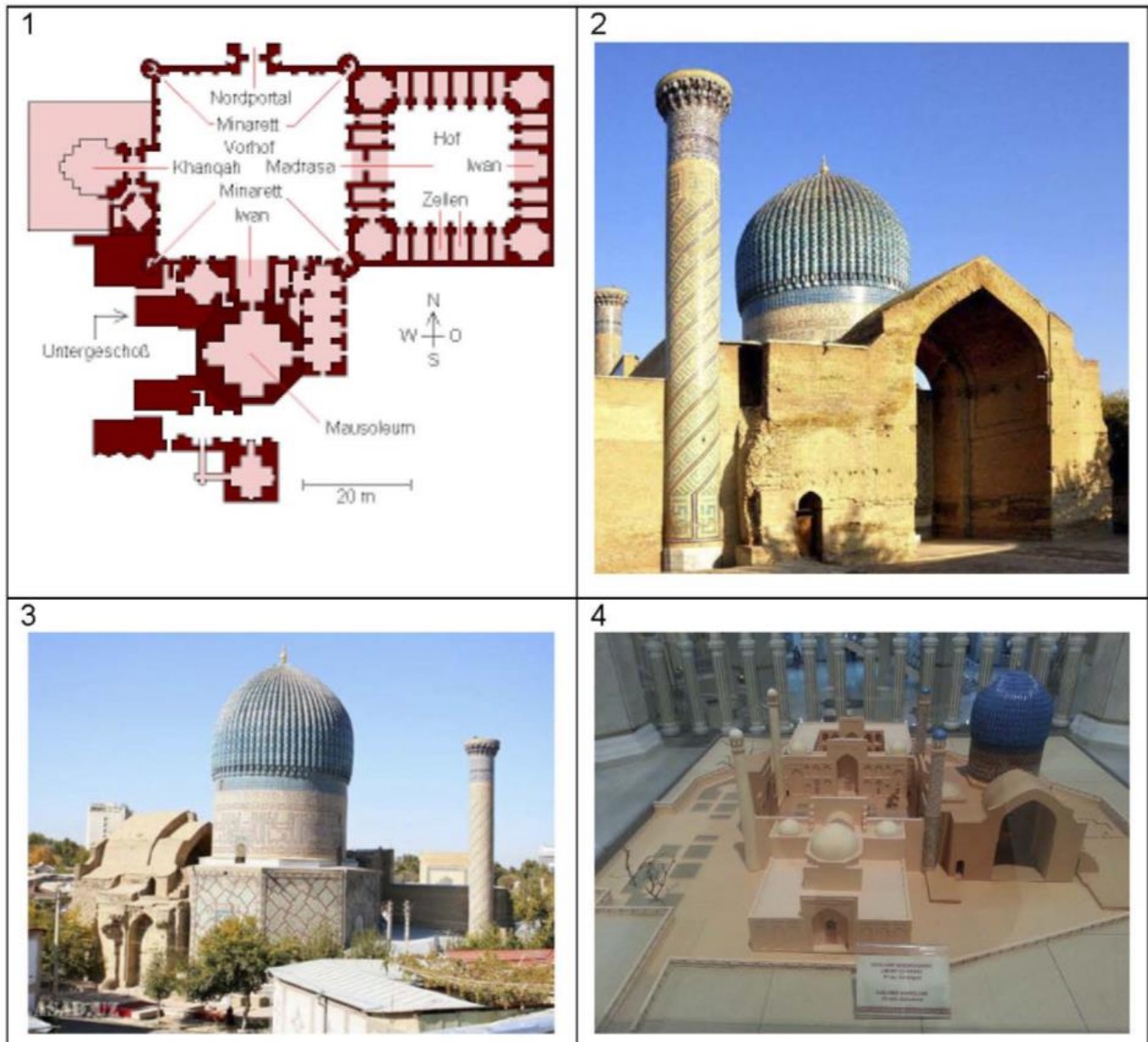

**Figure 7.** The Guri-Emir complex, Samarkand: 1—The plan (the reconstruction) [83]; 2, 3—The general view from the western side (beginning of the XXI century) [84,85]; 4—The reconstruction of the complex option (the model) [86].

The issue of the upper part of the minarets is debatable. In general, initially, minarets were independent structures. However, as noted by Sh. E. Ratiya [60] (p. 101), in the XIV century, for the first time in the architecture of Central Asia, there was a combination of two architectural forms—portal-domed buildings and minarets. Located symmetrically on both sides of the giant portal of the building, the minarets gave it the appearance of a complete, integral composition. Previously, minarets had only a functional religious purpose. Subsequently, during the period of the highest development of this type of portal structures, the minarets did not lose their functional-religious purpose, but acquired a new, architectural–tectonic significance. They became a characteristic part of the composition of structures. minarets are either directly adjacent to the portal or fixed to the corners of the extended facade.

Corner minarets, according to many researchers [4,10,50–52], had lanterns. This comes from the original function of minarets as observation towers. In addition to the comfort of staying on the observation deck of the guards, the lantern isolated the internal staircase from rain and snow. The confirmation of this hypothesis is in a comparison of detached medieval minarets and the corner minarets of one of the buildings of the late XVIII century. However, unlike Bukhara, where the lanterns have been preserved, in Samarkand, now almost all minarets are crowned only with a developed cornice. Additionally, this cornice, in fact, is the base of the lantern. There is a hypothesis about the destruction of lanterns in the process of historical development (earthquakes, deviation from the vertical due to subsidence of the soil). In the considered Guri-Emir complex, the crowning of minarets is difficult to determine—the painting by V. V. Vereshchagin (1870) shows a minaret with an almost completely destroyed top, on which only fragments of the "under-lantern" cornice are visible. The presented reconstruction option implies the presence of lanterns on both existing and restored minarets. The lanterns on the minarets in the Ulugbek Madrasah, which was built in 1415—1420 (39°39′17″ N, 66°58′28″ E), could have collapsed with a strong deviation of the minarets as a result of several earthquakes in the XIX century. Moreover, the most significant damage was created during the earthquakes of 1817/1818 and 1897. The minarets resisted, however, although they deviated. The largest deviation of the northeastern minaret was recorded in 1918. By the way, it is on this minaret that a low cylindrical superstructure above the "under-lantern" cornice has been preserved. This minaret was straightened in 1932 (engineers V. G. Shukhov, M. F. Mauer). The southeastern minaret was straightened in 1965 (engineers E. M. Handel, E. O. Nelly). Similar minarets in the Sher-Dor Madrasah opposite, which was built in 1619–1636 (39°39′17″ N, 66°58′34″ E), deviated slightly in the process of historical development. However, even in this building, the minarets do not have lanterns. The presence of an internal staircase implies the relatively active operation of these minarets. Accordingly, a lantern is needed to prevent rain from getting inside. Currently, the problem of insulation from rain is solved by a roof made of steel sheets. However, the question remains about the reasons for refusing to restore the lanterns. The complex restoration work carried out over many years included the straightening of existing minarets, the construction of destroyed minarets, and the restoration of cladding. However, the restoration of the lanterns crowning the minarets, which is a relatively simple event, was not carried out on any Samarkand monument of the Timurid era. Figure 8 shows modern roofs over Samarkand minarets.

The territory on which the city of Bukhara is located is much less susceptible to seismic impacts than Samarkand. This is one of the reasons for the significantly better preservation of medieval minarets in this region. Another factor is the significantly larger diameter of the trunk, which provides better stability for the building as well as powerful foundations with greater depth. The Kalyan minaret was built in 1127 in Bukhara (39°46′33″ N, 64°24′51″ E). The builder's name was Bako. The minaret has a height of 46.5 m. At the bottom, the trunk diameter is 9.7 m, and at the top—6.0 m. The rotunda lantern that completes the building has 16 through openings with a pointed arch. The upper part of the lantern has a multi-tiered cornice, above which there is a cone-like wedding, which is a continuation of the axial rod. The rotunda lantern was partially destroyed due to shelling in 1920. In 1925, the appearance of the minaret was restored. Earthquakes caused some damage to the stalactite cornices. These elements were restored in 1980. A complete restoration of the minaret was carried out in 1997. The Kalyan Minaret served as a prototype for similar buildings in the region. The earliest example of the imitation of this minaret is in the minaret in Vabkent (40°01′11″ N, 64°31′04″ E), which was built in 1198–1199. It has a height of 40.3 m. At the bottom, the trunk diameter is 6.2 m, and in the upper part—2.8 m. The rotunda lantern that completes the building has 8 through openings with a pointed arch. The diameter of the lantern is 3.66 m.

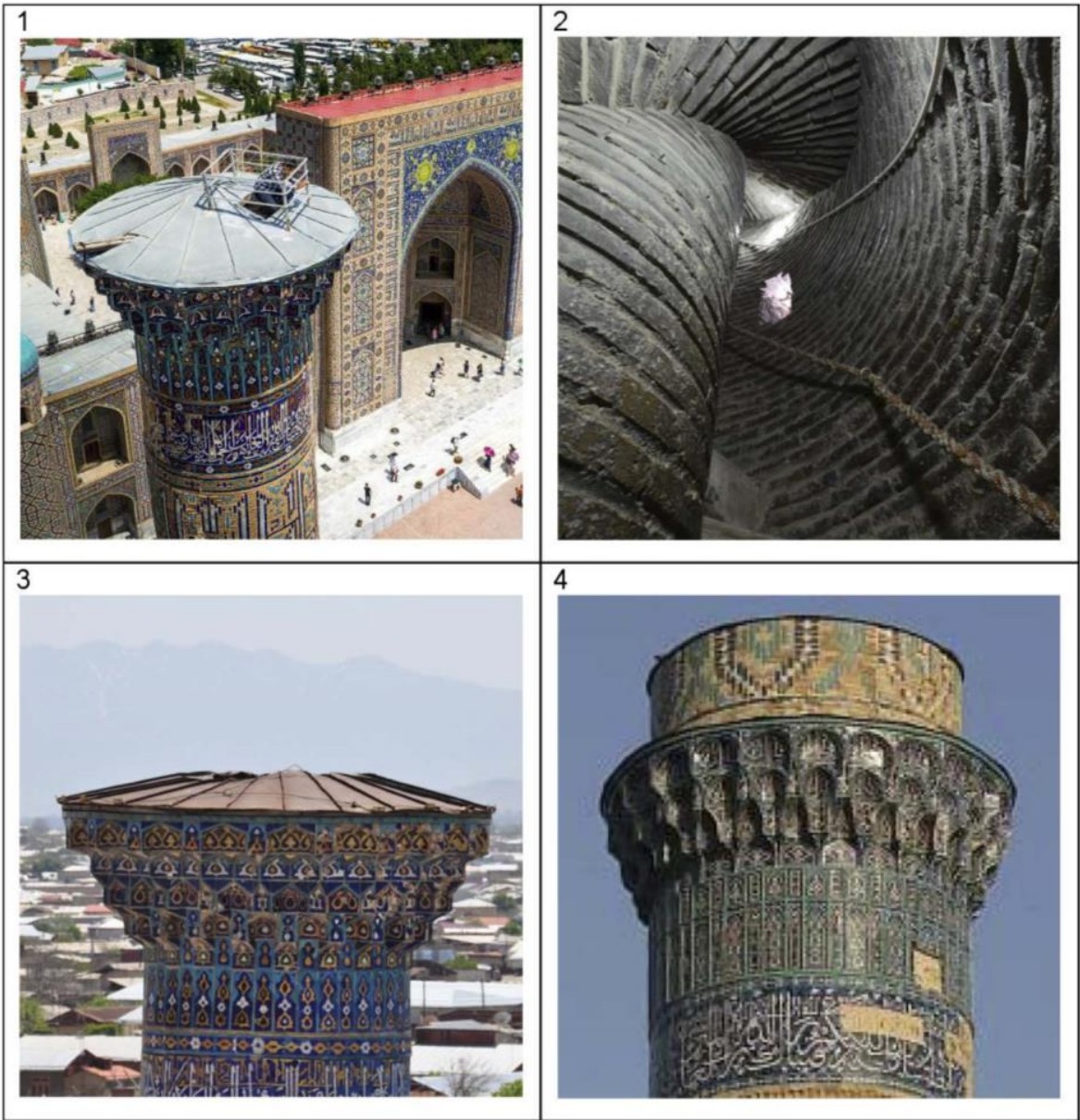

**Figure 8.** Medieval minarets of Samarkand: 1—The roof of the Ulugbek Madrasah minaret [87]; 2—The internal staircase in the minaret of the Ulugbek Madrasah [88]; 3—The roof of the Sherdor Madrasah minaret [89]; 4—The Bibi-Khanim mosque minaret top [90].

Corner minarets, designed in the medieval "Bukhara" style, are indicative in the Topchi-Bashi Madrasah in Bukhara. It was built in the second half of the XVIII century (its approximate coordinates are 39°46′14″ N, 64°25′73″ E). The building has corner minarets with lanterns, compositionally and plastically similar to medieval minarets in Bukhara and Vabkent. The building received significant damage as a result of shelling and was dismantled in the 1920s. The street named after Bahauddin Naqshbandi now runs through the place where the "Red" madrasah was located. The elegant plastic solution of the madrasah can be seen in the photo from S. M. Prokudin-Gorsky in 1911 [91,92]. This image, in some sources, mistakenly indicates the location of the object in Samarkand. In the context of the problem under consideration, the corner minarets are of interest. Their shape directly correlates with medieval minarets. The relatively thick trunk is crowned with a rotunda

lantern with six arched openings. The supports of the pointed arches form paired columns. A belt with ceramic lining is placed under the lantern. Unlike the blue and purple cladding in medieval analogues, green cladding is used here. The rotunda lantern has a developed multi-tiered cornice. The main facade of the madrasah itself has a three-part structure with an arched portal in the center. The corners of the portal are accentuated by small turrets with sharp peaks. These kinds of "pinnacles" are the crowning of three-quarter round pilasters. These free-standing and corner minarets are shown in Figure 9.

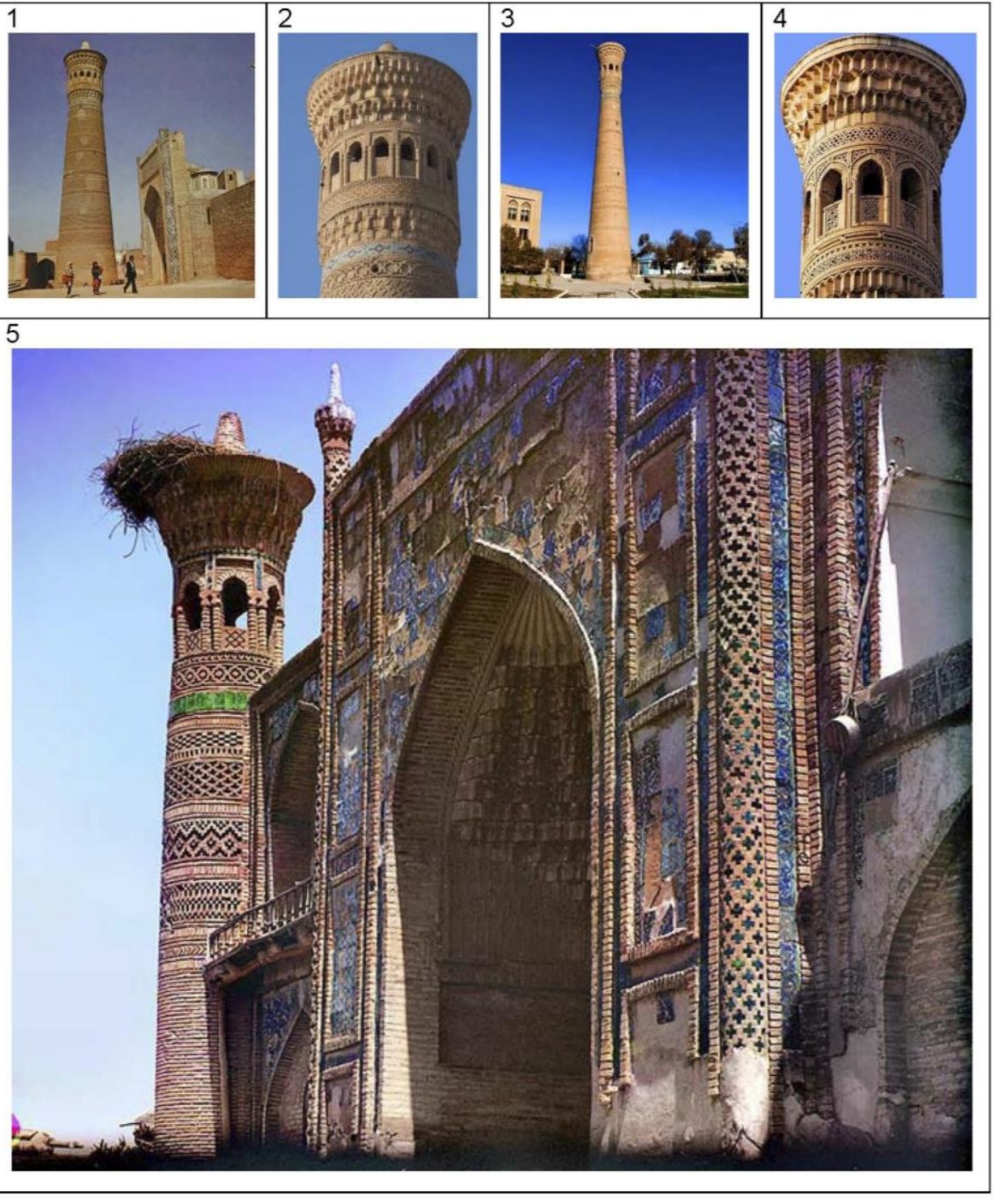

**Figure 9.** minarets of the Central Asia: 1, 2—The Kalyan minaret, Bukhara [93,94]; 3, 4—The minaret, Vabkent [95,96]; 5—The Topchi-Bashi Madrasah, Bukhara [97,98].

The Ak-Saray Palace complex was built in 1380–1404 in Shakhrisyabz (39°03′39″ N, 66°49′45″ E). The architect's name is Muhammad Yusuf Tabrizi. It is assumed that the height of the portal of the main entrance was 56.0 m. The arch of this portal was the largest in Central Asia—22.3 m. It is believed that the targeted destruction of the complex began in 1570 by order of the then ruler of Bukhara, who controlled the region. The brick gradually began to be pulled apart for the construction of other buildings. However, in 1707, the Bukhara ruler Ubaydullah Khan lived in the palace for some time. According to some reports, the portal collapsed in the first half of the XVIII century. To date, two dilapidated pylons have been preserved. Their height is about 38.0 m. The general view of the preserved part of the portal of the main entrance to the Ak-Saray complex and the variant of its reconstruction in the form of a layout are shown in Figure 10.

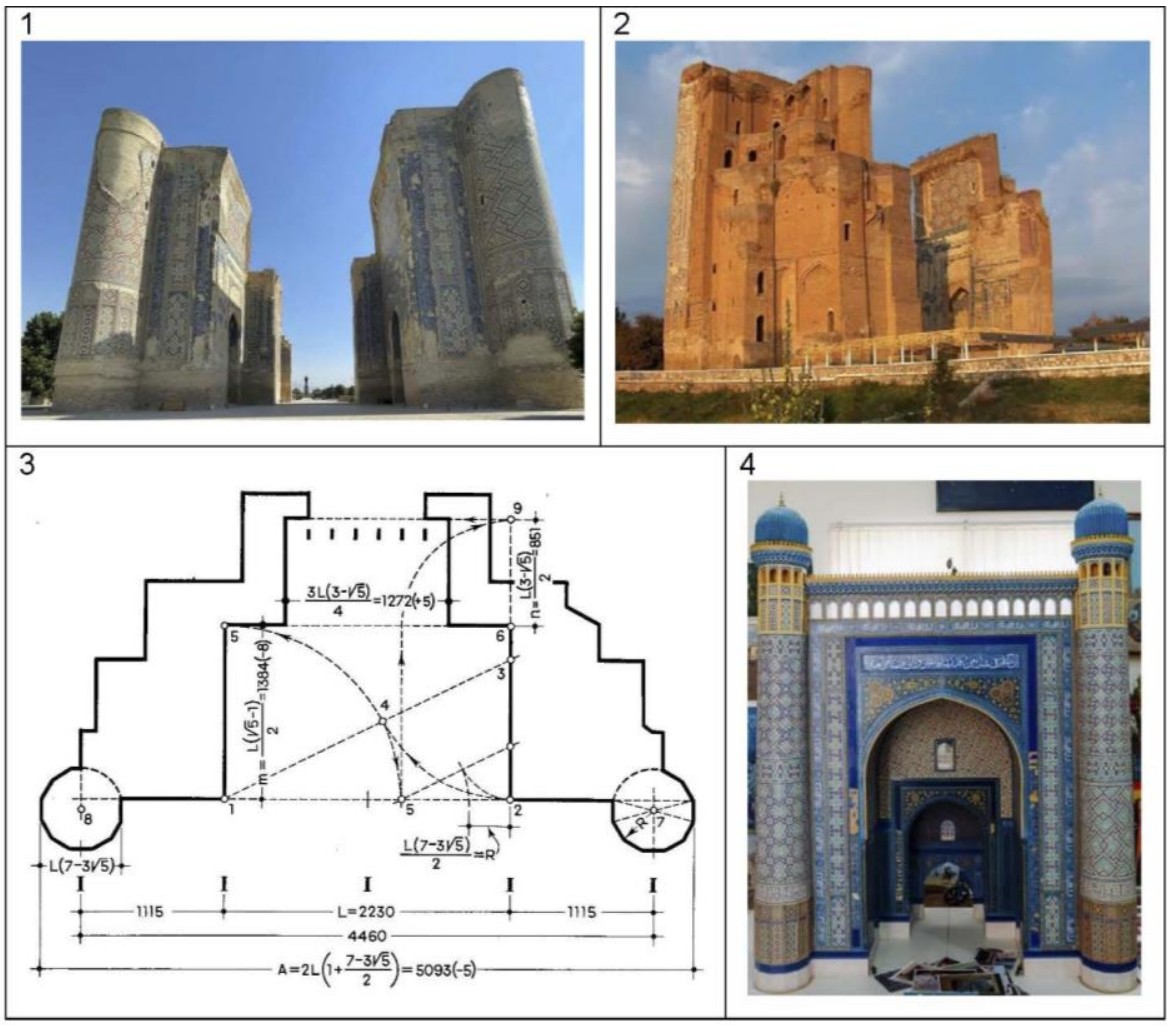

**Figure 10.** The Ak-Saray complex, Shakhrisyabz: 1, 2—The general view of the main entrance as of the beginning of the XXI century [99,100]; 3—The plan (the drawing—M. S. Bulatov) [59] (p. 166); 4—The reconstruction of the main entrance (the model) [101].

The Cathedral ("Jami") mosque Bibi-Khanim was built in 1399–1404 in Samarkand (39°39′38″ N, 66°58′45″ E). The complex was a rectangular courtyard 64 × 78 m enclosed by a colonnade of 480 columns. There was an entrance portal on the east side. Its height was 33.15 m. On the opposite side was the main mosque, and on the right and left (from the north and from the south) in the structure of the colonnade—small mosques. The corners of the complex were accentuated by minarets, which presumably had a height of 50.0 m. It is assumed that the minarets had a multi-tiered solution in the upper part. The

corner minarets of the entrance portal had a circular cross-section, and the minarets of the main mosque had a multifaceted one. The main facade of this grandiose structure was a harmonious combination of a giant majestic portal with minarets, behind which a huge dome towered. It is significant that this dome was completely obscured by the portal and was not visible from the side of the main facade. The minarets narrowed upwards. The planes of the portal walls and minarets are divided into separate false arches and narrow rectangular panels decorated with various geometric ornaments and inscriptions from mosaics and majolica. The flat ornamented archivolt of the portal arch rests on the capitals of round three-quarter columns mounted at the corners of the portal opening. The walls of the portal niche are entirely lined with geometric stylized ornaments of brick mosaic, which are interspersed with tiled rosettes of carved mosaic and majolica. The basement was lined with marble. The destruction of the complex began due to natural reasons at the beginning of the XV century when the dome of the mosque began to crack. A few years later, the arch of the entrance portal collapsed. The most serious damage occurred as a result of the earthquake of 1897. After this earthquake, the upper part of the preserved north-western minaret was dismantled. By the beginning of the twentieth century, the colonnade had almost disappeared. The rest of the buildings have been preserved in varying degrees of destruction. The preserved minaret has a low parapet above the lantern cornice (Figure 8). Major restoration work began in 1968 and it continues to this day. Figure 11 shows: the complex preserved by the middle of the twentieth century; the plan of the complex; the geometric analysis of the composition of the plan of the main mosque; and the current state of the main mosque as a result of reconstruction. The sequence of construction of similar buildings of the Timur era: the Ak-Sarai Palace, Shakhrisyabz; the Mausoleum-Khanaka of Ahmed Yassawi, Turkestan; The main mosque of Bibi-Khanim, Samarkand—are shown in Figure 12. The reconstruction of the proposed view of the main mosque of Bibi Khanum in Samarkand is shown in Figure 13.

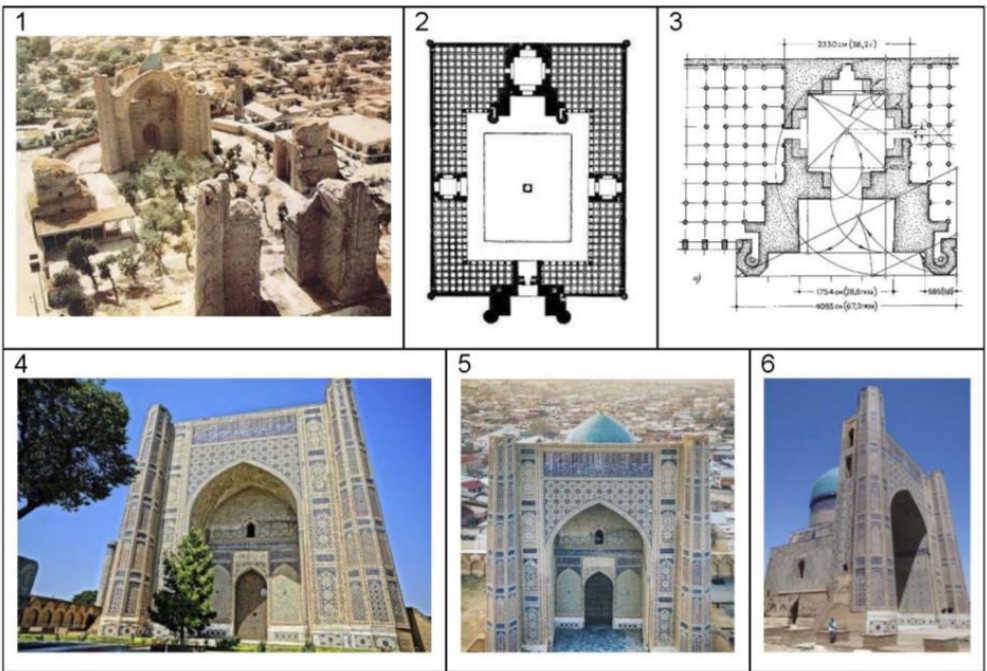

**Figure 11.** The main mosque of Bibi-Khanim, Samarkand: 1—The complex preserved by the middle of the twentieth century (photo from 1968) [102]; 2—The plan (the reconstruction) [103]; 3—The geometric analysis of the composition of the plan (M. S. Bulatov) [59] (p. 173); 4, 5, 6—The reconstruction of the late twentieth century [104–106].

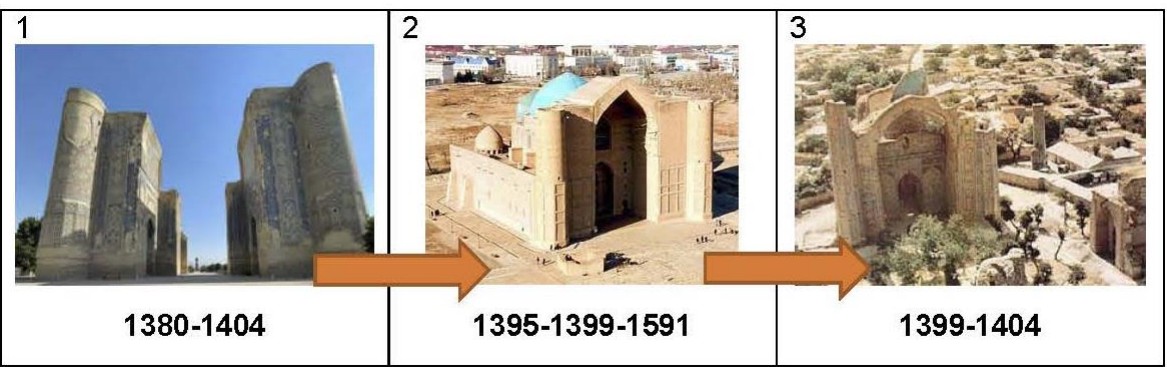

**Figure 12.** The sequence of construction of similar buildings of the Timur era (state at the beginning of the last quarter of the twentieth century): 1—Ak-Sarai Palace, Shakhrisyabz [99]; 2—Mausoleum-khanaka Ahmed Yassavi, Turkestan [80]; 3—The main mosque Bibi-Khanim, Samarkand [102].

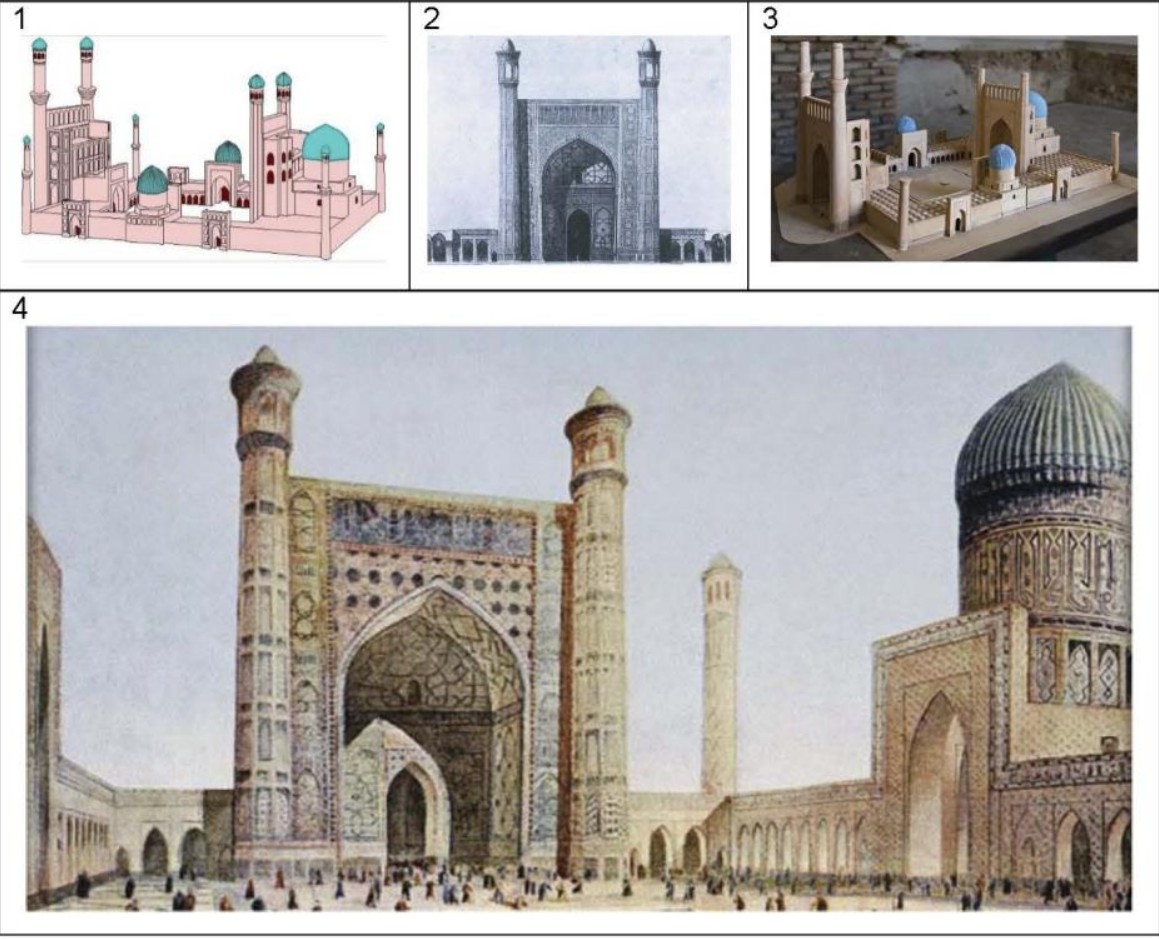

**Figure 13.** The reconstruction of the proposed view of the Bibi-Khanim main mosque in Samarkand: 1—The general view drawing [107]; 2—The drawing (Sh. E. Ratiya) [60] (p. 87); 3—The model [108]; 4—The general view (Sh. E. Ratiya) [60], (Figure 77).

The architectural composition of the buildings under consideration serves as the basis for the analysis carried out further in order to determine the unrealized original plan of the Turkestan mausoleum-khanaka.

### 3. Results and Discussion

In general, the idea of restoring the original condition of the building is methodologically based on reconstruction by analogy, the measure of quality of which is the criterion of reliability. Moreover, we are not talking here about restoration in kind, but only about conditional reconstruction in the drawing. In the case of the mausoleum-khanaka, this can serve as a starting point for discussion and stimulate further research on this issue. Moreover, despite the long-term study of the monument by many researchers, the issue of restoring the original design concept of the mausoleum-khanaka is being emphasized for the first time.

The following parameters are the initial data for the reconstruction of the proposed appearance of the south-eastern facade of the mausoleum-khanaka of Ahmed Yassawi:

(1)    the width of the facade along the axes of the corner minarets—45.00 m;
(2)    width of the north-western façade—45.96 m;
(3)    the width of the facade in the dimensions of the corner minarets—49.9 m;
(4)    the width of the base of the corner minarets is 6.9 m

The closest analogue of the portal, according to many researchers, is the portal of the main mosque of the Bibi-Khanim complex. It is even assumed that the construction of the mosque was led by craftsmen from the same corporation that previously built the mausoleum-khanaka [60] (p. 96). In the compositional analysis of Sh. E. Ratiya [60] (p. 92), the width of the facade of the main Bibi-Khanim mosque is 40.88 m. It consists of seven parts: two minarets (5.84 × 2), two piers (5.84 × 2), and an arched opening (5.84 × 3). Accordingly, it is believed that the height of the portal could be 40.88 m (based on the assumption of the same width and height). The height of the corner minarets to the cornice is 51.16 m (10.28 m above the portal). In the analysis of M. S. Bulatov [59] (p. 173), the width of the facade of the main Bibi-Khanim mosque is 40.85 m, and the corner minarets are 5.85 m wide. The assumption about the similarity of the width and height of the portal is based on the analysis of the existing building of the Ulugbek Madrasah mosque, built in 1415—1420 in Samarkand (39°39′17″ N, 66°58′28″ E). In this building, the width and height of the portal is 31.80 m (M.S.Bulatov [59] (p. 195)). These data are given in the drawings of Sh. E. Ratiya and M. S. Bulatov, shown in Figure 14. If we assume that this technique is suitable for calculating the height of the portal of the mausoleum-khanaka of Ahmed Yassawi, then it should be 49.9 m. The height of the minarets could be equal to $A2^{1/2}$ = 49.9 × 1.41 = 70.36 m. The drawing of this construction is shown in Figure 15. However, the assumption about the equality of the width and height of the portal comes from the analysis of the Ulugbek Madrasah. Additionally, it was built later. The portal of this building has no minarets (they are located on the edges of the facade). Accordingly, the width of the facade, which affects the height, can be considered the size of the northwest facade of the mausoleum-khanaka. The width of this facade is 45.9 m. Thus, the height of the portal could also be 45.9 m. The minarets towered above the portal to a height of triple thickness (6.9 × 3 = 20.7 m). The total height of the minarets could be 66.6 m. The drawing of this construction is shown in Figure 16.

G. A. Pugachenkova points out: "Timur's monumental buildings are interpreted as a volume closed from the external environment. The law of iron symmetry reigns in everything. The facade composition is based on a simple, clear idea of a huge portal arch, with powerful corner towers flanking the walls. The static equilibrium of the peshtak array is not disturbed by a certain vertical orientation of both arches and towers, which is ordered by a system of typical divisions, in the form of framing rectangular frames. At the same time, the architect uses the same compositional scheme both in the cathedral mosque of Samarkand, and in the Ak-Sarai Palace, and in the mausoleum of the cleric Hazret Ahmed Yasawi, and in the dynastic tomb of Dorus-Siadat" [74] (p. 70). Schematically, this is shown in Figure 17.

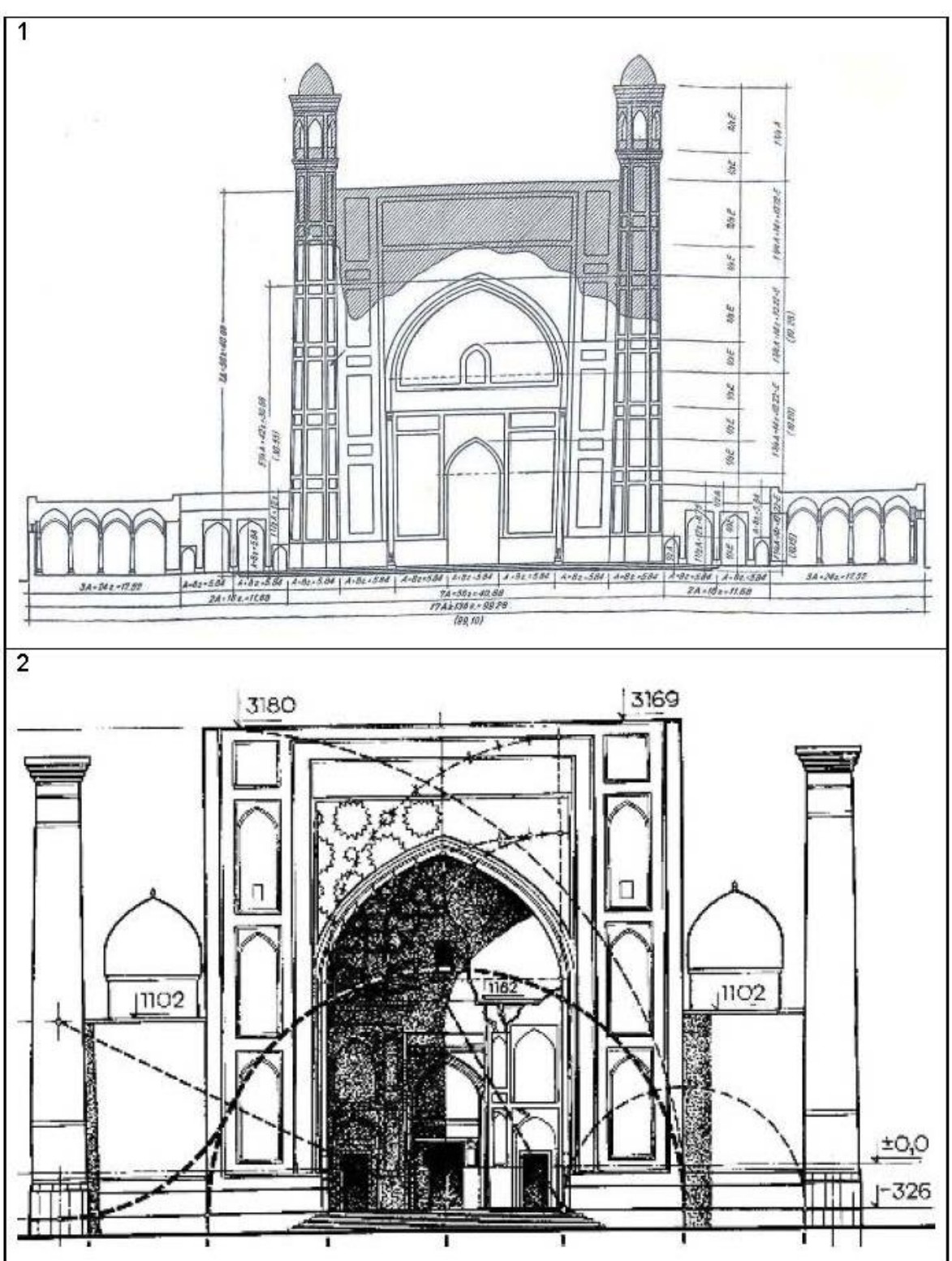

**Figure 14.** Geometric analysis of the composition of the facades of medieval buildings: 1—The Bibi-Khanim main mosque in Samarkand, the alleged view of the main facade (Sh. E. Ratiya) [60] (p. 92); 2—The Ulugbek Madrasah in Samarkand, geometric analysis of the composition of the facade (M. S. Bulatov) [59] (p. 195).

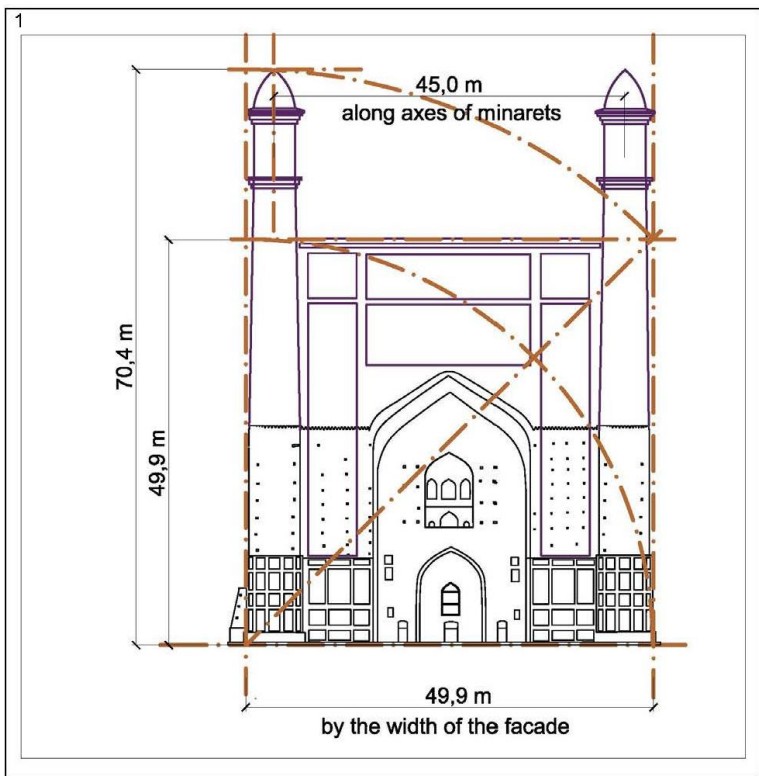

**Figure 15.** The portal of the mausoleum-khanaka of Ahmed Yassawi by analogy with the Bibi-Khanim main mosque in Samarkand (the drawing from the authors).

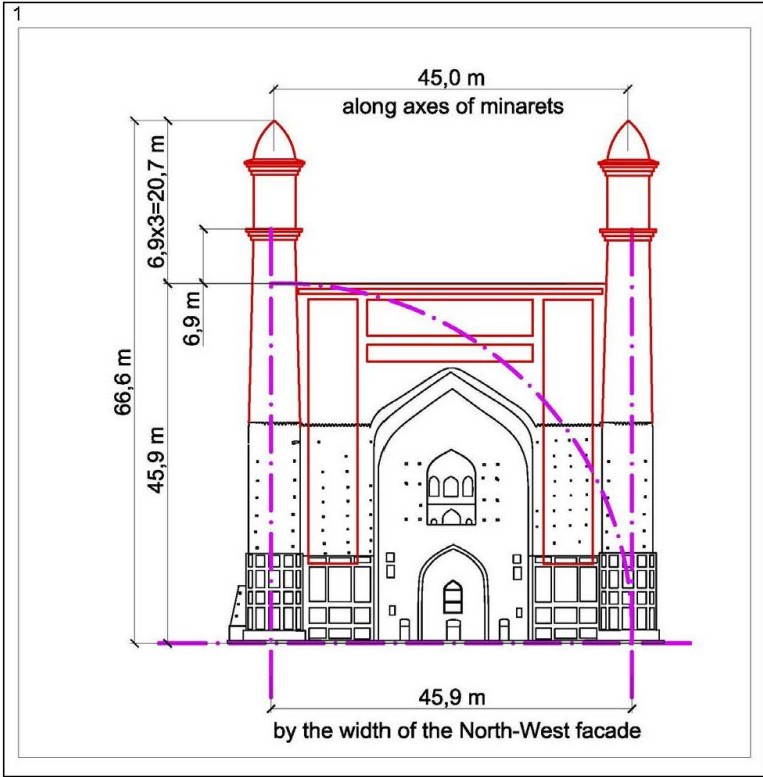

**Figure 16.** The portal of the mausoleum-khanaka of Ahmed Yassawi by analogy with the Ulugbek Madrasah in Samarkand (the drawing from the authors).

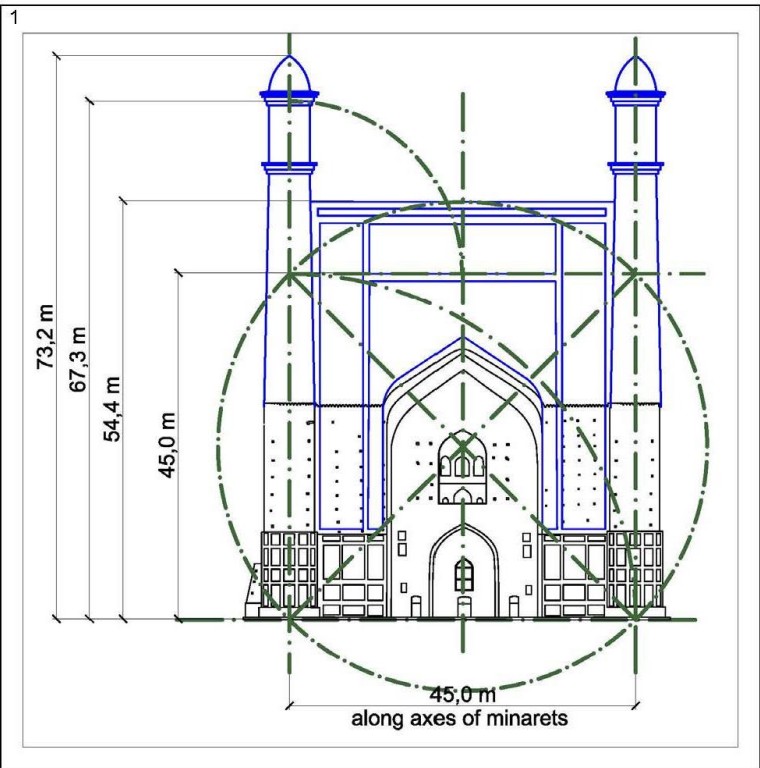

**Figure 17.** The portal of the mausoleum-khanaka of Ahmed Yassawi by analogy with the portal of the main entrance to the Ak-Sarai Palace in Shakhrisyabz (the drawing from the authors).

The appearance of the mausoleum-khanaka, as M. E. Masson pointed out [4] (p. 14), has similarities with the main entrance to the Ak-Saray complex in Shakhrisabz. The most similar are the angular minarets, which, having a polygonal lower part, turn into a cylinder above. "The wonderful scenery of the ruins of Aksaray and the surviving skeleton of the portal of the mausoleum of Khoja Ahmed in comparison complement each other in mental reconstruction. In order to restore the appearance that the peshtak of the Turkestan monument should have, it is necessary to imagine, instead of the ugly battlements of the portal walls, a rectangular frame embracing the castle of the arch brought down under Abdullakhan, which is now restlessly piercing into the air" [4] (p. 14).

In the compositional analysis of M. S. Bulatov [59] (p. 166), the portal of the Ak-Sarai Palace has a width of 50.93 m in the dimensions of the corner minarets, and the width along the axes of the minarets is 44.60 m. If we assume that the height of the portal was determined by its width, then the portal could be 50.93 m high.

However, as mentioned earlier, some studies indicate an estimated portal height of 56.0 m. This value is obtained if we use the well-known proportionation formula $A5^{1/2}$:2 = 1.118A, that is, 50.93×1.118 = 56.94 m. If we apply this formula to the mausoleum-khanaka of Ahmed Yassawi, then the height of its portal could be 49.9×1.118 = 55.79 m.

Approximately such a value can be obtained if the following geometric construction is applied. For the base size "A", take not the width of the facade, but the axes of the minarets. Since ancient times, and still yet in most cases, the marking of the dimensions of the building for construction was carried out along the axes of the main elements. It is fair to assume that in the era under consideration (XIV—XV centuries) this technologically simple and reliable technique could be used. The fact is that it is difficult to determine the actual dimensions of buildings (with an accuracy of centimeters, and sometimes decimeters) before the start of construction, since the thickness of the cladding is often unknown and errors are possible when performing foundational and masonry work (until now, the construction standards of some countries indicate the parameters of permissible deviations from the project up or down). Accordingly, it is technologically more rational to set the

dimensions of the building on the construction site while focusing on the center axes. Based on this, it is possible to carry out a geometric construction of the facade based on the size of 45.0 m, corresponding to the axes of the minarets. The height of the portal is determined by the size of the upper part of the circle described around the square of the axes of the minarets. This value is approximately 54.4 m. If we assume the presence of crowning teeth (they are shown, among other things, on the reconstruction of the portal of the Ak-Sarai Palace—Figure 10), then an approximate value of 56.0 m is obtained. This is the size obtained on the basis of calculations by analogy with the estimated height of the portal of the Ak-Sarai Palace. minarets could have a height of 1.5 A to the cornice (67.3 m). Additionally, the total height of the minarets with domes could be 73.2 m (the domes have the size of the width of the base of the minaret). The drawing of this construction is shown in Figure 15.

Based on these geometric constructions, it can be suggested schematically how the facades of the mausoleum-khanaka of Ahmed Yassawi could look if the original plan had been fully implemented. This is shown in Figure 18. However, looking at these facades, the question arises about the height of the main dome. Unlike the other domes (above the mausoleum, above the mosque—Figure 4), which are double, this dome has a single solution. The profile of this dome is similar to the profiles of the inner domes. In addition, unlike most domes of the Timur era, the main dome of the mausoleum-khanaka has a low drum. It may have been planned later, upon completion of construction, to increase the height of the drum and make an additional external dome. At the same time, the top of the dome could be at the height of the portal (about 50 m—about 8–10 m higher than the preserved dome)—for example, the dome in the main mosque of the Bibi-Khanim complex was made. An approximate scheme of the possible solution of the main dome in comparison with analogues is shown in Figure 19. Its design is comparable to that of the dome of the Guri-Emir mausoleum (in the Turkestan mausoleum-khanaka, the dome diameter is about 18 m; in the Samarkand mausoleum, the dome diameter is about 15 m) and with the dome of the main mosque of the Bibi-Khanim complex (this dome is slightly smaller than in the mausoleum-khanaka). As indicated, likely due to the desire of the customer to complete the construction of the mausoleum-khanaka later, this partially explains the incompleteness of the arch and the entire portal. At the same time, the preserved main dome, hypothetically, was faced from the outside in order to preserve the building for a possibly long break in construction. Similar to the question of the shape of the portal, the question of the shape of the main dome of the mausoleum-khanaka requires additional study. Variants of reconstruction of similar buildings of the Timur era: Ak-Sarai Palace, Shakhrisabz (A. A. Akhmedov, 1980–2021); mausoleum-khanaka of Ahmed Yassawi, Turkestan (K. I. Samoilov, 2022); the main mosque Bibi-Khanim, Samarkand (S. E. Ratiya, 1950) are shown in Figure 20.

If, as a basis for constructing a possible scheme of decorative cladding, we take the examples discussed earlier (the entrance portal of the Ak-Saray Palace complex in Shakhrisyabz, the main mosque of the Bibi-Khanim complex in Samarkand, the Ulugbek Madrasah in Samarkand), then the portal of the Khanaka mausoleum could look as shown in Figure 21. As in the main volume of the building of the mausoleum-khanaka, the decoration of the portal could be as follows. The bottom of the walls is lined with smooth slabs of yellow sandstone. A wide strip of about 1.5 m with edges of blue samples could be located above it. The strip itself could have been laid out with a complex geometric ornament. The main element of the drawing could be an eight-ray star and alternating figured slabs of yellow sandstone and painted majolica. A narrow ribbon of a new row of sand tiles could serve as a border, above which all of the free space of the walls could be occupied by patterns of white and blue tiles on a yellow background of facing bricks. Some of the patterns could represent epigraphic decor.

Naturally, the study of recreating the original design of the general appearance of the mausoleum-khanaka is only part of the problem. It is necessary to conduct a significant

amount of research in various fields. Determining the historically adequate general appearance of the building is the initial stage. Among the issues are determining the value and necessity of preserving historical layers; the condition of the main structures, walls, floors and the need to strengthen them (the foundations were reinforced with reinforced concrete during restoration); the use of historically adequate or new building materials and structures; the possibility of seasonal or year–round work; and much more.

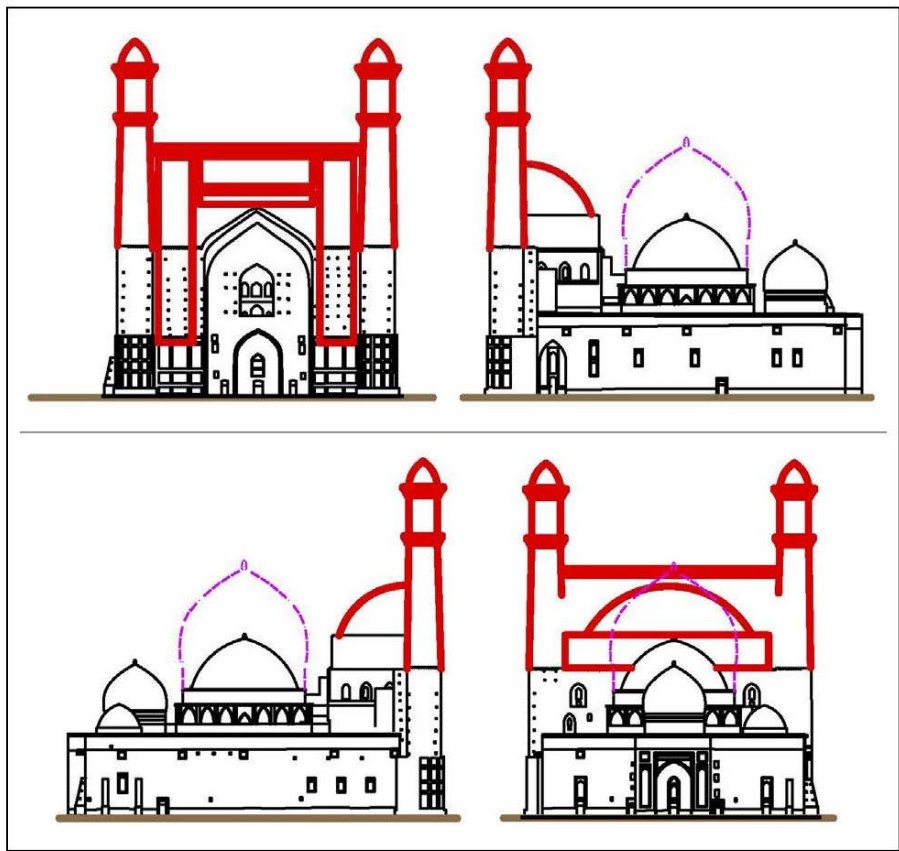

**Figure 18.** The alleged schematic view of the mausoleum-khanaka of Ahmed Yassawi in Turkestan facades—the alleged view of the portal is shown in red, the alleged view of the main dome is shown in purple (the drawing from the authors).

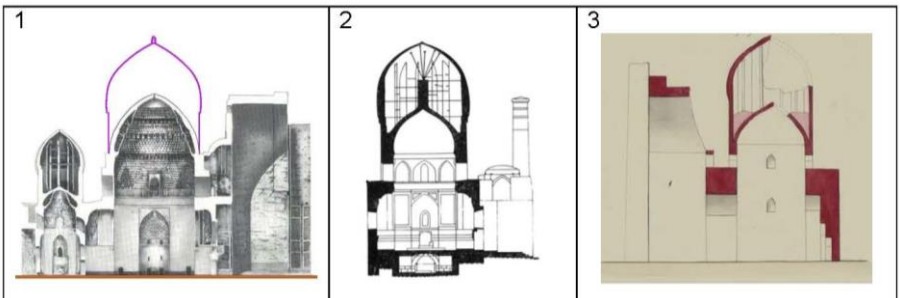

**Figure 19.** The assumed shape of the outer dome over the central part of the mausoleum-khanaka of Ahmed Yassawi in Turkestan (1 [109] + the drawing from the authors) in comparison with analogues—the Guri-Emir mausoleum (2 [110]) and the main mosque of Bibi-Khanim complex (3 [111]) in Samarkand.

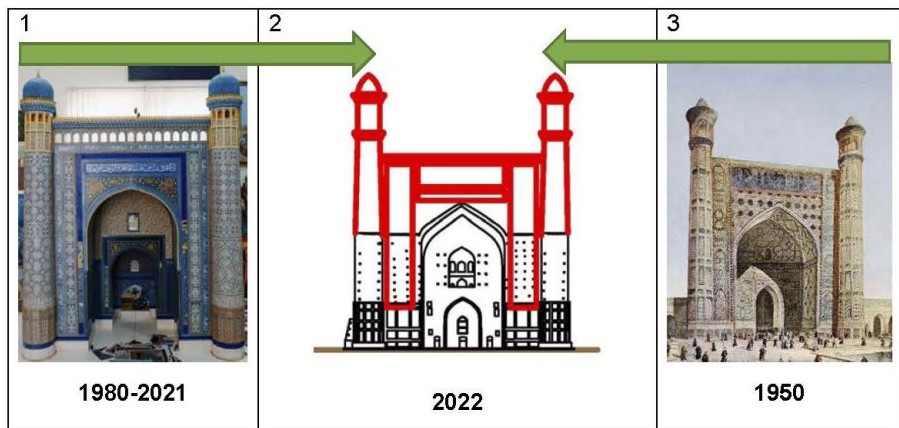

**Figure 20.** Reconstruction options for similar buildings of the Timur era: 1—Ak-Sarai Palace, Shakhrisyabz (A. A. Akhmedov, 1980–2021 [101]); 2—Mausoleum-khanaka Ahmed Yassavi, Turkestan (K. I. Samoilov, 2022—drawing from the authors); 3—The main mosque Bibi-Khanim, Samarkand (Sh. E. Ratiya, 1950 [58], (Figure 77)).

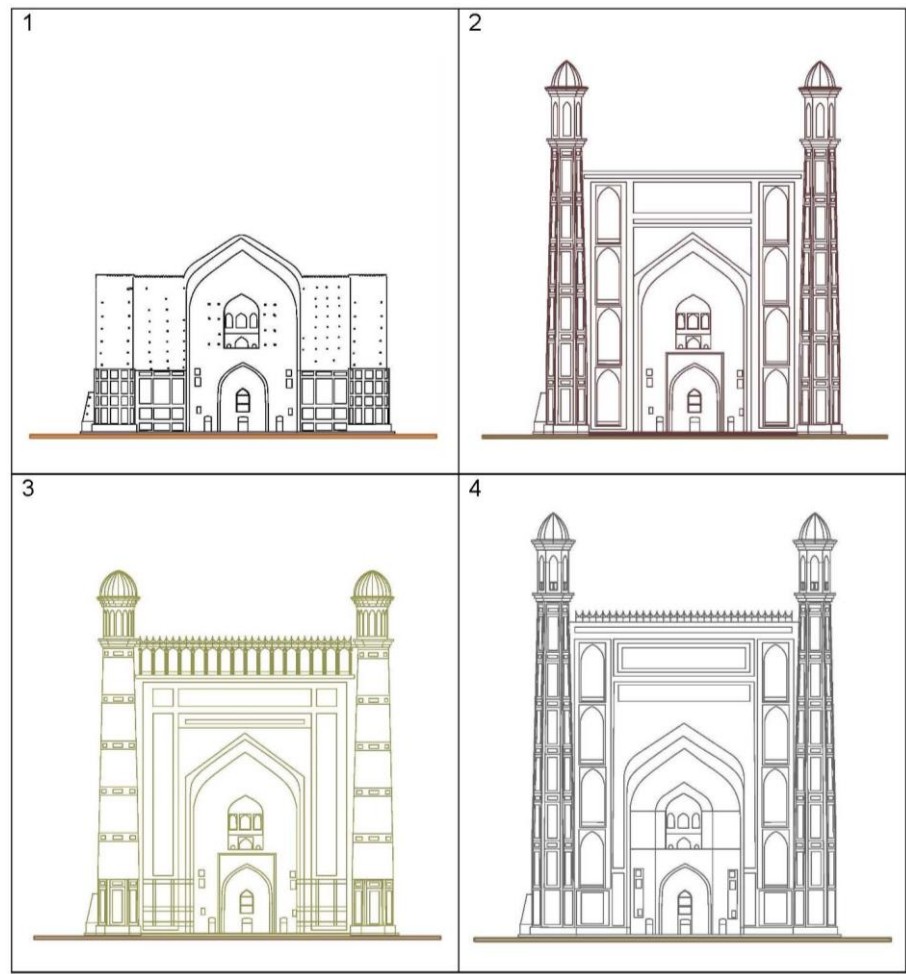

**Figure 21.** Views of the portal of the mausoleum-khanaka of Ahmed Yassawi in Turkestan (the drawing from the authors): 1—the current situation at the beginning of the XXI century; 2, 3, 4—the proposed types of the portal based on the interpretation of the techniques of decoration of the Ulugbek Madrasah in Samarkand, the main entrance to the Ak-Sarai Palace in Shakhrisyabz and the Bibi-Khanim main mosque in Samarkand.

In the Turkestan mausoleum-khan aka, it is necessary to determine the expediency of preserving "historical scars". These are the deflection of the right archivolt of the portal arch, and the reinforcing buttresses on the southwest and northwest facades. When the foundations are reinforced under the entire building, the buttresses cease to have a functional purpose. It is also advisable to redo the portal arch, since the archivolt deflection is a blatant deformation. It arose most likely due to a non-compliance with the masonry technology at its completion in the XVI century. The presence of modern steel cable ties shows the emergency condition of this structure. The construction of the portal to the design height will be one of the ways to strengthen this arch. Accordingly, the implementation of the original idea from a purely speculative problem passes into the category of a structurally necessary measure to preserve the strength of the architectural monument.

The next question is to determine the feasibility of using "historical" bricks produced in Sairam for possible completion. The bricks were laid on an alabaster mortar. Given the presence of cladding, the expediency of using historical materials for masonry walls is debatable. It is possible to use modern wall materials. In principle, it is possible to strengthen the walls with steel frames with steel grids. These elements will be hidden by the cladding. Accordingly, the appearance of the building, directly perceived by visitors, will not depend on the main structural materials.

When the building is completed, it will, in a sense, lose its historicity. However, since the mausoleum-khanaka of Ahmed Yassawi in Turkestan is now considered as the most important part of the historical and cultural heritage of Kazakhstan, the completion of the construction of this building will be an excellent illustration of the continuity of culture and the greatness of the deeds of Kazakhstan's ancestors. This is especially important if we recall the famous saying of Emir Timur, recorded on the gates of the Ak-Sarai Palace complex in Shakhrisyabz: "If you doubt our power, then look at our buildings".

The discussion about the prospect of the rapid completion of the mausoleum-khanaka is still unproductive, as extensive additional research is still needed. However, the potential of the history of the construction of this complex encourages the continuation of work:

(1)     In the twelfth century, the first mausoleum was erected;
(2)     By the middle of the thirteenth–early-fourteenth centuries, it became part of a complex of buildings with an open courtyard and a well in the middle;
(3)     The end of the fourteenth century united all of the buildings and the courtyard into one building with a large dome and a majestic portal (the portal arch and the outer dome remained unfinished);
(4)     At the end of the sixteenth century, the arch of the portal was completed, and the upper part of the unfinished portal and corner towers were strengthened;
(5)     In the next two centuries, constant repairs protected the building from destruction;
(6)     In the second half of the nineteenth century, repair and restoration work intensified, and the cladding of the main dome was started, which allowed it to be protected from destruction (the absence of an external dome became critical for the building);
(7)     At the end of the nineteenth–mid-twentieth century, the building was completely cleared of later additions and almost all of the cladding was restored (with the exception of the portal);
(8)     At the turn of the twentieth and twenty-first centuries, repair and restoration work was actively carried out.

Each of these stages was accompanied by a study of the complex (at a scientific and methodological level accessible for each period). Now, having conducted extensive research and public discussion, it is advisable to continue the work of the ancestors and complete the construction in memory of them, realizing their grandiose plan. Moreover, there are some examples of such solutions in the region.

Of course, we can leave the building in its "historical form" while continuing conservation and restoration work. However, research on how the khanaka mausoleum might look if it was completed at the beginning of the XV century is, in itself, of some specific scientific and general cultural interest. They are connected with the detailing of the forms

of the unfinished portal, the features of the missing external dome on the high drum in the compositional center of the mausoleum-khanaka, and the specifics of the drawings of the cladding of these elements. At this new stage, detailed drawings and 3D models can be made.

## 4. Conclusions

Supporters of the complete restoration of monuments want to see them today in their former beauty and grace, making them available for tourist displays and the possibility of using them for museumification or adaptation for other purposes. Opponents of full restoration advocate for the minimization of restoration work, since, in this case, the monuments will lose their historicity, and then the historical and artistic value will be lost. On the one hand, there is the principle of holistic restoration and the idea of restoring the ruins in their original state for the sake of integrity of aesthetic perception. This principle is historically associated with the restorations of E. Viollet-le-Duc. On the other hand, there is the principle of conservation (in fact, anti-restoration). This is associated with the name of J. Ruskin. The "original state" is considered to be the ruin itself. The bearer of historical value is not the integral image lost and restored in the work of imagination, but the materiality of fragments preserved from antiquity. Both of these principles interact dialectically. There is no restoration without conservation intervention—without strengthening the structure and removing irreversible damage. There are no conservation projects that are completely free from the need to restore ruins.

Examples of unfinished constructions are a separate problem. In this case, it is sometimes advisable to recognize the need for the completion of construction. Numerous examples demonstrate the successful execution of these works. Two methods are usually used. One determines the completion of construction in other forms corresponding to a new view of architectural and artistic shaping. This method has historical value as it shows the dynamics of changing stylistic preferences. The second option seems interesting from a historical point of view. This is the completion of construction in forms presumably corresponding to the original plan.

In any case, for the remaining unfinished or dilapidated buildings, considered as important milestones in the development of the region, it is advisable to completely either "physically" or "virtually" restore them. Whether or not its continuity is considered by the State as fundamental is one of the features of the continuity of culture. The preservation of heritage in the form in which it has reached in the modern era is also very promising. It gives you the opportunity to feel the historical depth and the scale of times. The importance of this is confirmed by the romantic attention that periodically arises in the culture to the "Aesthetics of ruins".

**Author Contributions:** Conceptualization, K.S., B.K., G.S. and A.A.; methodology, K.S., B.K. and G.S.; software, K.S. and A.A.; validation, K.S., G.S. and A.A.; formal analysis, K.S., B.K., G.S.; investigation, K.S., B.K., G.S. and A.A.; resources, K.S., B.K. and A.A.; data curation, B.K. and K.S.; writing—original draft preparation, K.S., B.K., G.S. and A.A.; writing—review and editing, K.S., B.K. and G.S.; visualization, A.A. and K.S.; supervision, K.S. and G.S.; project administration, K.S. and B.K.; funding acquisition, K.S. and A.A. All authors have read and agreed to the published version of the manuscript.

**Funding:** This research received no external funding.

**Data Availability Statement:** Not applicable.

**Acknowledgments:** Authors express their gratitude to the Google search engine, the USA Library of Congress, and the British Library, whose resources are used for the selection of theoretical material.

**Conflicts of Interest:** The authors declare no conflict of interest.

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
