# Peer review of "The Version of the Composition of the Mausoleum-Khanaka Khoja Ahmed Yassawi Main Facade in Turkestan"

_heritage, doi:10.3390/heritage6020074_

Round 1

Reviewer 1 Report (Previous Reviewer 1)

This version of the paper seems more complete and above all clearer than the previous one. There are more references and the authors have better contextualized some methodological choices of the research.

However I do not agree with the authors when they choose to only summarize the studies/state of the art in the introduction. I believe that only the most important contributions should be selected to frame the issue; these should be critically exposed by the authors. Citing a long list of scholars and indicating the contents of each in a superficial way is of no help to the reader and indeed makes it difficult to understand.

Author Response

Dear Reviewer,

Many thanks for the detailed analysis of our manuscript “The version of the composition of the mausoleum-khan aka Khoja Ahmed Yassawi main facade in Turkestan”.

Taking into account your instructions, we have supplemented the text with a description of the largest and most important works that have formed a modern understanding of the problem of the development of the Khoja Ahmed Yasawi complex in Turkestan.

They are – one of the first published descriptions made by M.S.Bekchurin in 1860; detailed studies by M.E.Masson 1929-1930 (it was M.E.Masson who first suggested the similarity of architectural forms of the unfinished portal of the mausoleum-khanaka in Turkestan and the portal of the main entrance to the Ak-Sarai Palace in Shakhrisyabz); completed in 1950 G.I.Patsevich analysis of repair and restoration works carried out at the site in 1939-1941; comprehensive studies of the monument carried out by L.Yu.Mankovskaya in the 1960s and 1970s (the monument was first studied at the dissertation level, and architectural and archaeological research allowed L.Yu.Mankovskaya to formulate a number of important postulates on typological features of the entire region). In the late 1980s, B.T.Tuyakbayeva described the specifics of the mausoleum's epigraphic decoration at the dissertation level. These works are the basis of all other studies, the total number of which is now approaching a hundred.

We considered it necessary, for reasons of scientific ethics, to preserve in the manuscript references to some of the most important, from our point of view, studies that affect certain aspects of the mausoleum-khanaka complex and similar structures in the region. It was the analysis of the totality of these studies that allowed us to put forward a hypothesis about the design appearance of the main portal and the main dome.

Thanks again for your comments.

With respect,

Konstantin Samoilov,

Bolat Kuspangaliyev,

Gaukhar Sadvokasova,

Aizhan Akhmedova

Reviewer 2 Report (Previous Reviewer 2)

The changes are now sufficient to publish

Author Response

Dear Reviewer,

Many thanks for the analysis of our manuscript “The version of the composition of the mausoleum-khan aka Khoja Ahmed Yassawi main facade in Turkestan”.

We have supplemented the text with a description of the largest and most important works that have formed a modern understanding of the problem of the development of the Khoja Ahmed Yasawi complex in Turkestan.

They are – one of the first published descriptions made by M.S.Bekchurin in 1860; detailed studies by M.E.Masson 1929-1930 (it was M.E.Masson who first suggested the similarity of architectural forms of the unfinished portal of the mausoleum-khanaka in Turkestan and the portal of the main entrance to the Ak-Sarai Palace in Shakhrisyabz); completed in 1950 G.I.Patsevich analysis of repair and restoration works carried out at the site in 1939-1941; comprehensive studies of the monument carried out by L.Yu.Mankovskaya in the 1960s and 1970s (the monument was first studied at the dissertation level, and architectural and archaeological research allowed L.Yu.Mankovskaya to formulate a number of important postulates on typological features of the entire region). In the late 1980s, B.T.Tuyakbayeva described the specifics of the mausoleum's epigraphic decoration at the dissertation level. These works are the basis of all other studies, the total number of which is now approaching a hundred.

We considered it necessary, for reasons of scientific ethics, to preserve in the manuscript references to some of the most important, from our point of view, studies that affect certain aspects of the mausoleum-khanaka complex and similar structures in the region. It was the analysis of the totality of these studies that allowed us to put forward a hypothesis about the design appearance of the main portal and the main dome.

Thanks again for your comments.

With respect,

Konstantin Samoilov,

Bolat Kuspangaliyev,

Gaukhar Sadvokasova,

Aizhan Akhmedova

This manuscript is a resubmission of an earlier submission. The following is a list of the peer review reports and author responses from that submission.

Round 1

Reviewer 1 Report

The paper deals with a very complex issue from a historical-architectural and conservation-restoration point of view. The proposed case study (the mausoleum-khanaka Khoja Ahmed Yassawi in Turkestan) is very interesting and inspiring.

Knowledge of the monument is clear to the authors but the paper proposes a series of studies by other researchers listed uncritically according to a long sequence of bibliographical references (up to line 66). The authors should select these studies and critically propose to the readers what they share about these and what instead appears to them to be of little significance or importance for the purposes of the research. The discussion of the previous conservation and restoration interventions that have involved the monument is instead very clear and exhaustive, even if some historical photos of the interventions would be desirable and would accompany readers in understanding the theme.

The authors also propose an articulated examination of similar case studies for construction techniques and/or construction period and/or geographical area. However they assume such cases in a comparative process that does not take into account the risks and dangers of the so-called "analogical criterion": those examples can provide interesting indications but there is always a margin of variation, of uniqueness, specific to each building and each monument; it is not clear how the authors have taken this aspect into account and this should be specified as it is a fundamental issue.

The goal is the graphic restitution/reconstruction of the main facade, which is certainly "of some specific scientific and general cultural interest". The absence of references to any virtual reconstructions based on this process is unusual, especially nowadays. The theme of "virtual restoration" seems to be suspended in the field of research. Similarly, the authors' position towards the different conservation/restoration attitudes is not clear, since the graphic reconstruction does not exhaust the discussion and does not provide an answer to the question that they themselves posed in the paper (on the legitimacy of restoration interventions on unfinished monuments).

Finally there are some formatting issues:

- why don't the bibliographic references follow the numerical order in the text?

- the captions of images 6 and 10 are on the following pages;

- scientific discourse would require a greater fluidity of the text, for example by eliminating the numerous subordinate sentences (“the fact that …”, “it is necessary to …”).

For all these reasons a major review seems absolutely necessary.

Author Response

Dear reviewer,

Thank you so much for Your attention to our manuscript “The version of the composition of the mausoleum-khanaka Khoja Ahmed Yassawi main facade in Turkestan”.

In the new edition of the manuscript, we have tried to take into account all Your comments:

- the review of the literature briefly indicates the essence of the research that we relied on in our work;

- the mausoleum-khanaka from the 16th century, when the arch of the main portal was completed, was subjected exclusively to repair and restoration work (especially in the second half of the 19th century: the device of four buttresses, the beginning of the cladding of the main dome; by the middle of the 20th century, most of the wall cladding collapsed; it was restored by the end of the 20th century; along with this, works on strengthening walls and domes; reinforced concrete foundation was made). The condition of the building in the middle of the 20th century is shown in Figure 2. The restored cladding is shown in Figure 4. It was not possible to find photos of laying the foundations for the building and photo-fixing of other repairs in the public domain. There are several photographs from 1911, one of which is shown in Figure 3.

- we realize that the proposed reconstruction of the completion of the portal and the construction of the outer dome is based on the principle of analogy. However, many researchers, for example, G.A.Pugachenkova (quote on page 32 – source 40, p.70) point to the uniformity of compositional techniques of the Timur era. Accordingly, we assumed that the analog approach is quite appropriate. Figure 10 shows successively erected monuments from 1380 to 1404 . One customer, similar dimensions make it possible to assume an analogy.

- virtual reconstructions of the mausoleum-khanaka Ahmed Yassawi in Turkestan could not be found. In conversations with fellow restorers, it turned out that such a task has not yet been set. Our proposed option is most likely the first work in this direction. Naturally, it is necessary to conduct very in-depth research on this issue. After that, it will be possible to create high-quality three-dimensional images. Our proposal for the completion of the portal and the dome can be the beginning of a scientific discussion. At the same time, the publication of this material in the respected “Heritage” magazine will help this.

- the legality of the completion of the portal and the construction of the dome is based (in our opinion) on the history of the construction of this complex: the mausoleum of the 12th century - buildings around the courtyard with a well of the 13th century – the construction of the portal and dome over the courtyard of the 14th century – the completion of the portal arch of the 16th century – repair and restoration work of the second half of the 19th and 20th centuries. In all periods there were funds for performing certain types of work. Now, due to the unfinished portal, the right part of the arch is in critical condition (this is shown by the arrow in Figures 2 and 4). We consider it legitimate to consider the issue of continuing construction (realizing the idea of the ancestors). Moreover, the mausoleum-hakaka has a huge spiritual significance for modern Kazakhstan.

- bibliographic references are in alphabetical order – it seems to us that this is convenient from the point of view of finding books and articles;

- we have tried to partially correct the wording, bringing them closer to modern English;

- we also corrected the formatting.

If you think that You need to clarify any other positions, then we are ready to do it.

Thank You again for your attention to our manuscript. We hope that the corrections we have made will allow You to recommend our manuscript for publication in the “Heritage” journal.

With respect,

Konstantin Samoilov,

Bolat Kuspangaliyev,

Gaukhar Sadvokasova,

Aizhan Akhmedova

Author Response

Dear reviewer,

Thank you so much for Your attention to our manuscript “The version of the composition of the mausoleum-khanaka Khoja Ahmed Yassawi main facade in Turkestan”.

In the new edition of the manuscript, we have tried to take into account all Your comments:

- the review of the literature briefly indicates the essence of the research that we relied on in our work;

- the mausoleum-khanaka from the 16th century, when the arch of the main portal was completed, was subjected exclusively to repair and restoration work (especially in the second half of the 19th century: the device of four buttresses, the beginning of the cladding of the main dome; by the middle of the 20th century, most of the wall cladding collapsed; it was restored by the end of the 20th century; along with this, works on strengthening walls and domes; reinforced concrete foundation was made). The condition of the building in the middle of the 20th century is shown in Figure 2. The restored cladding is shown in Figure 4. It was not possible to find photos of laying the foundations for the building and photo-fixing of other repairs in the public domain. There are several photographs from 1911, one of which is shown in Figure 3.

- we realize that the proposed reconstruction of the completion of the portal and the construction of the outer dome is based on the principle of analogy. However, many researchers, for example, G.A.Pugachenkova (quote on page 32 – source 40, p.70) point to the uniformity of compositional techniques of the Timur era. Accordingly, we assumed that the analog approach is quite appropriate. Figure 10 shows successively erected monuments from 1380 to 1404 . One customer, similar dimensions make it possible to assume an analogy.

- the authors who made certain proposals are indicated in the appropriate places;

- the purpose of the manuscript is to draw attention to the problem of unfinished buildings, which is important for the modern culture of Kazakhstan;

- the examples considered at the beginning – the Karakhan mausoleum in Taraz, the Aisha-bibi mausoleum in the village of Aisha bibi – demonstrate the fundamental possibility of partial alteration or restoration of historically valuable buildings in Kazakhstan; in neighboring Uzbekistan, the Main Bi-bi-Khanim mosque has been almost completely restored from ruins.

- the added figures 10 and 18 show the presence of similar buildings in the region during the Timur period.

- the legality of the completion of the portal and the construction of the dome is based (in our opinion) on the history of the construction of this complex: the mausoleum of the 12th century - buildings around the courtyard with a well of the 13th century – the construction of the portal and dome over the courtyard of the 14th century – the completion of the portal arch of the 16th century – repair and restoration work of the second half of the 19th and 20th centuries. In all periods there were funds for performing certain types of work. Now, due to the unfinished portal, the right part of the arch is in critical condition (this is shown by the arrow in Figures 2 and 4). We consider it legitimate to consider the issue of continuing construction (realizing the idea of the ancestors). Moreover, the mausoleum-khanaka has great spiritual significance for modern Kazakhstan.

We hope that our article will serve as a starting point for a scientific discussion. This is especially interesting, since before that the task of completing the portal and erecting the dome was not set.

If you think that You need to clarify any other positions, then we are ready to do it.

Thank You again for your attention to our manuscript. We hope that the corrections we have made will allow You to recommend our manuscript for publication in the “Heritage” journal.

With respect,

Konstantin Samoilov,

Bolat Kuspangaliyev,

Gaukhar Sadvokasova,

Aizhan Akhmedova

Round 2

Reviewer 1 Report

The new version of the paper undoubtedly clarifies some aspects highlighted in the previous review and resolves some ambiguities. However, it requires further clarifications relating to the methodological setting since the previous restorations of buildings in the same geographical context or whose first phases date back to the same historical period are not sufficient to define the choice of the monument as methodologically adequate for a basic reason: each building is a case in itself.

Nor is it enough to recall the two great poles of a scientific and disciplinary debate - by now also very historicized and outdated - which see the two main references in Viollet-le-Duc and Ruskin. In between there are many considerations, even recent ones, that should be taken into consideration. The principle of analogy should not be rejected a priori, but it should be adequately motivated, otherwise the authors risk falling into the misunderstandings of the so-called historical restoration of the early 20th century.

I therefore suggest to the authors to take a further step forward to declare why they do not include additions with openly new and contemporary parts (with a contemporary architectonical language): as already mentioned, this is a legitimate choice but one that needs to be discussed more fully.

Furthermore, in the Introduction I strongly advise against indicating only in brackets the main points of view of the numerous scholars cited. Their arguments should be explained more fully, if appropriate, and not as a mere list; I believe, however, that the authors should make a narrower selection of these studies, as some of them are perhaps superfluous or redundant with respect to the considerations already expressed.

Author Response

Dear reviewer,

Thank you so much for Your attention to our manuscript “The version of the composition of the mausoleum-khanaka Khoja Ahmed Yassawi main facade in Turkestan”.

We have tried to take into account your comments and have supplemented the manuscript.

The analysis methodology used is based on the principle of analogy.

The principle of analogy is justified in this case, since there is a triad – one historically short period, one region, one customer. It is difficult to assume that in this combination of factors, a cardinal difference in the principles of shaping is possible. The historical restoration of the beginning of the last century was based on the fantasy of the "ideal forms" of the epoch. In the situation under consideration, it is proposed to use standard shaping techniques that have been repeatedly implemented in buildings of similar size and function.

At the beginning of the study, examples of construction completion in forms different from the original ones are mentioned. This "modernization" is seen by the authors as inadequate for the object under consideration. The proposed option suggests the possibility of reproducing the original design of the mausoleum-khanaka Ahmed Yassawi. This approach is seen by the authors as the most attractive from a historical and educational point of view. It shows the greatness of the ancestors' plan. This is very important for the modern outlook of the country's citizens. This demonstrates the deep historical roots of the country's modern culture.

The use of modern architectonic language in this case seems unacceptable to the authors. The goal is not just to complete the construction, the goal is to reproduce the original design.

In accordance with Your recommendation, information about some of the most important studies has been supplemented. Authors of the manuscript did not set the task of studying the history of the monument research. Therefore, the studies mentioned in the literature review reflect the main part of the spectrum of available works on the topic under consideration. The works of the largest experts on the topic (Masson, Man’kovskaya, Pugachenkova, Zasypkin and other), published in the first half of the twentieth century, and modern studies summarizing and clarifying the existing body of knowledge are mentioned here. For scientific and ethical reasons, the authors want to keep references to all these works. The presented set of studies is interesting in itself. It shows the history of studying the problem. This is important for understanding the specifics of the topic. Brief descriptions of the conducted studies are given because they are of auxiliary importance for the topic under consideration. The question of the original appearance of the mausoleum-khanaka of Ahmed Yasawi has not yet been raised at the scientific and theoretical level. Our manuscript is the first on this topic. We hope that it will serve as the beginning of a fruitful discussion. Therefore, special attention is paid to similar buildings. They are seen as an opportunity to understand the original idea of the mausoleum-khanaka.

Thank You again for your attention to our manuscript. We hope that the corrections we have made will allow You to recommend our manuscript for publication in the “Heritage” journal.

With respect,

Konstantin Samoilov,

Bolat Kuspangaliyev,

Gaukhar Sadvokasova,

Aizhan Akhmedova
